# Mechanistic basis for PYROXD1-mediated protection of the human tRNA ligase complex against oxidative inactivation

Luuk Loeff[1], Alena Kroupova[1,4], Igor Asanović[2], Franziska M. Boneberg[1], Moritz M. Pfleiderer [1], Luca Riermeier [3], Alexander Leitner [3], Andrè Ferdigg [2], Fabian Ackle [1], Javier Martinez [2] & Martin Jinek [1] ✉

The metazoan tRNA ligase complex (tRNA-LC) has essential roles in tRNA biogenesis and unfolded protein response. Its catalytic subunit RTCB contains a conserved active-site cysteine that is susceptible to metal ion-induced oxidative inactivation. The flavin-containing oxidoreductase PYROXD1 preserves the activity of human tRNA-LC in a NAD(P)H-dependent manner, but its protective mechanism remains elusive. Here, we report a cryogenic electron microscopic structure of the human RTCB–PYROXD1 complex, revealing that PYROXD1 directly interacts with the catalytic center of RTCB through its carboxy-terminal tail. NAD(P)H binding and FAD reduction allosterically control PYROXD1 activity and RTCB recruitment, while reoxidation of PYROXD1 enables timed release of RTCB. PYROXD1 interaction is mutually exclusive with Archease-mediated RTCB guanylylation, and guanylylated RTCB is intrinsically protected from oxidative inactivation. Together, these findings provide a mechanistic framework for the protective function of PYROXD1 that maintains the activity of the tRNA-LC under aerobic conditions.

RNA molecules are present in all domains of life and undergo extensive post-transcriptional processing. The precursor transcripts of a subset of tRNA genes contain introns that are removed through a non-canonical spliceosome-independent pathway[1,2]. Non-canonical RNA splicing of precursor-tRNAs is a two-step enzymatic process that requires endonucleolytic excision of a single intron by the tRNA splicing endonuclease complex[3–5], after which the two tRNA exon halves are ligated by the tRNA-LC[6,7]. The core of human tRNA-LC is composed of the catalytic subunit RTCB and accessory proteins DDX1, CGI-99, FAM98B and ASHWIN[6,8,9]. In addition, the complex relies on the co-factor Archease to catalyze the guanylylation of RTCB[8,10]. Apart from its essential function in tRNA biogenesis, tRNA-LC is required for XBP1 mRNA splicing in the unfolded protein response[11] and has been implicated in RNA trafficking[12] and repair[13].

RNA ligation by RTCB enzymes proceeds through three distinct nucleotidyl transfer steps. In the first step, catalyzed by the Archease co-factor, a conserved histidine residue in the catalytic center is covalently linked to guanine-5′-monophosphate (GMP) through a nucleophilic attack on the α-phosphate moiety of guanosine triphosphate (GTP), releasing inorganic pyrophosphate[14–17]. Subsequently, the 2′,3′-cyclic phosphate of the 5′-exon half is hydrolyzed to yield a 3′-phosphate that executes a second nucleophilic attack on the phosphate group of the RTCB–GMP adduct, resulting in an activated RNA-(3′)pp(5′)G intermediate[14–17]. In the final nucleotidyl transfer step, the 5′-hydroxyl group of the 3′-exon half attacks the activated RNA-(3′) pp(5′)G intermediate of the 5′-exon half, resulting in ligation of the tRNA exon halves and release of GMP[16,17].

Human RTCB contains a strictly conserved catalytic center that comprises a cysteine, an aspartate and four histidine residues that coordinate two divalent metal ions that are essential for the ligation reaction, making RTCB a binuclear metalloenzyme[6,9,16]. The reactive cysteine residue in the catalytic center renders RTCB sensitive to

[1]Department of Biochemistry, University of Zurich, Zurich, Switzerland. [2]Max Perutz Labs, Medical University of Vienna, Vienna BioCenter, Vienna, Austria. [3]Institute of Molecular Systems Biology, Department of Biology, ETH Zürich, Zurich, Switzerland. [4]Present address: Centre for Targeted Protein Degradation, University of Dundee, Dundee, UK. ✉e-mail: jinek@bioc.uzh.ch

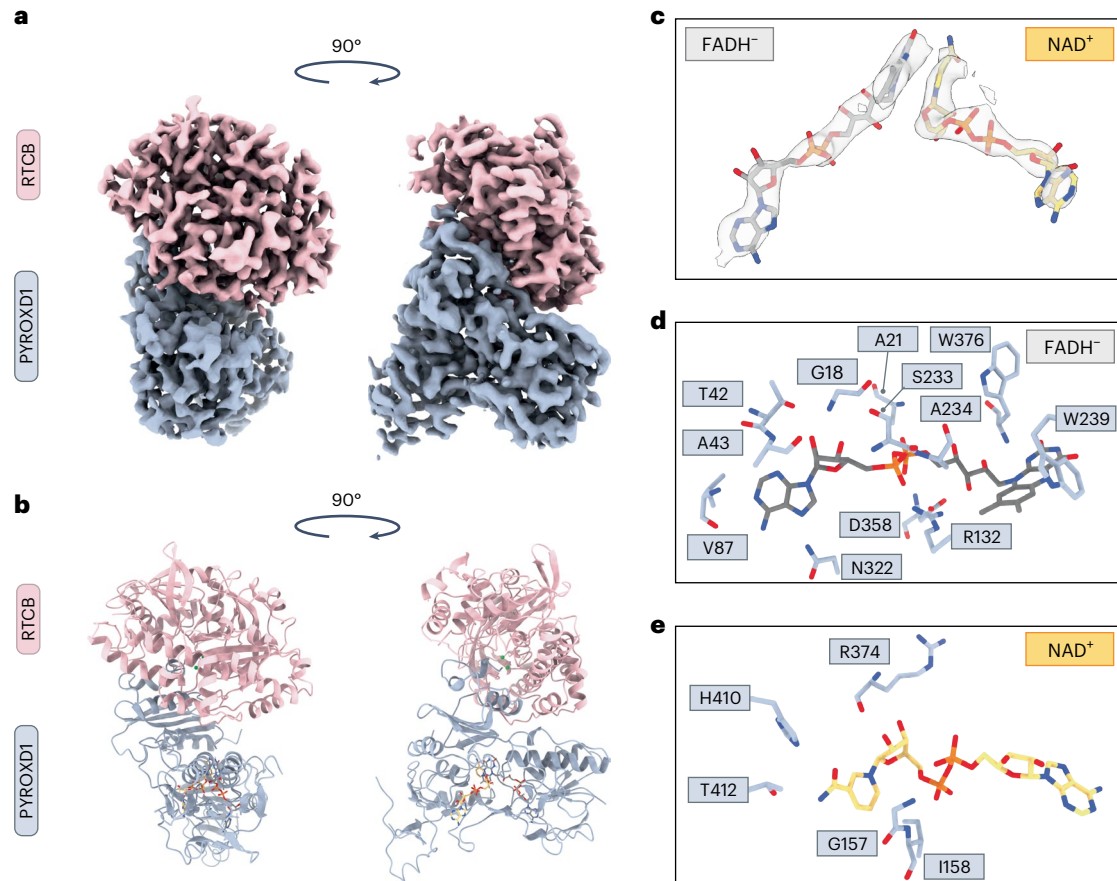

**Fig. 1 | Cryo-EM structure of the human RTCB–PYROXD1 complex. a**, Cryo-EM density map of the RTCB–PYROXD1 complex. **b**, Structural model of the RTCB–PYROXD1 complex. **c**, Zoom-in view of the nucleotide ligands in the RTCB–PYROXD1 complex, depicted in the redox states inferred from biochemical data (FADH⁻ in gray and NAD⁺ in orange). Map (light gray) contoured at σ = 10. **d**, Interaction map of residues within a 3.35 Å distance from the FADH⁻ ligand in the RTCB–PYROXD1 complex. **e**, Interaction map of residues within a 3.35 Å distance from the NAD⁺ ligand in the RTCB–PYROXD1 complex.

oxidative inactivation in the presence of copper(II) ions[18,19]. To preserve the function of RTCB under aerobic conditions, the tRNA-LC relies on the flavin adenine dinucleotide (FAD)-containing oxidoreductase PYROXD1 to protect its catalytic center from oxidation[19]. In vitro, FAD-bound PYROXD1 directly interacts with RTCB in a NAD(P)H-dependent manner[19], sustaining the ligase activity of the tRNA-LC under oxidizing conditions. Depletion of PYROXD1 in cells inhibits RTCB activity, resulting in the accumulation of unspliced tRNA exon halves and a decrease in XBP1 mRNA splicing during the unfolded protein response[19]. The presence of mutations in PYROXD1 has been shown to lead to myopathy[20] and a connective tissue disorder[21], highlighting the significance of these genetic variations for both muscular and connective tissue function. Despite the importance of PYROXD1 for sustaining RTCB activity in vivo, the mechanistic basis for this protective function has remained elusive.

Here, we present a cryogenic electron microscopy (cryo-EM) structure of human RTCB in complex with PYROXD1 and reveal the molecular basis for PYROXD1-mediated protection of RTCB. Our results demonstrate that the C-terminal tail of PYROXD1 directly interacts with the catalytic center of RTCB and reveal a NAD(P)H-dependent conformational switch in PYROXD1 that allosterically modulates the interaction with RTCB. We further show that guanylylation of the catalytic center of RTCB by Archease inhibits the recruitment of PYROXD1 and intrinsically protects RTCB from oxidative inactivation. Our results collectively reveal how PYROXD1 protects the catalytic subunit of the tRNA-LC from inactivation by reactive oxygen species.

## Results

### Molecular architecture of the RTCB–PYROXD1 complex

To elucidate the mechanism underlying the protective function of the flavoprotein PYROXD1 in preventing oxidative inactivation of RTCB, we first used affinity co-precipitation experiments to establish the optimal conditions under which RTCB and PYROXD1 stably interact. Co-precipitation of recombinant PYROXD1 by immobilized RTCB required the presence of 0.5 mM NADH (Extended Data Fig. 1a), in agreement with our prior work demonstrating that the interaction of RTCB and PYROXD1 is NAD(P)H-dependent and that NAD(P)⁺ does not support PYROXD1–RTCB complex formation[19]. Furthermore, PYROXD1 co-precipitation was observed in the presence of a variety of divalent metal cations (Ca²⁺, Co²⁺, Cu²⁺, Mg²⁺, Mn²⁺, Ni²⁺ and Zn²⁺), while chelation of divalent cations with EDTA abolished the interaction between PYROXD1 and RTCB (Extended Data Fig. 1b), confirming that the interaction is divalent cation-dependent[19]. As monitored by absorption spectroscopy, PYROXD1 catalyzed multiple-turnover oxidation of NADH to NAD⁺ (Extended Data Fig. 1c); the observed kinetics were consistent with a previous study showing that PYROXD1 catalyzes the initial formation of a NAD⁺–FADH⁻ charge-transfer complex (CTC), followed by slower reoxidation of the CTC by molecular O₂ (ref. 19).

PYROXD1 is able to oxidize both NADH and NADPH with comparable kinetics[19], although the physiological substrate is likely to be NADPH, given the nucleocytoplasmic localization of PYROXD1 (ref. 20). To visualize the interaction between RTCB and PYROXD1, we reconstituted the RTCB–PYROXD1 complex in the presence of NADH and Mg²⁺ ions and analyzed it using single-particle cryo-EM.

To overcome the orientation bias of the particles, two datasets were acquired using samples prepared in the presence and absence of octyl-beta-glucoside (Extended Data Fig. 2a) and combined to yield a molecular reconstruction of the RTCB–PYROXD1 complex at a resolution of 3.3 Å (Fig. 1a,b, Table 1 and Extended Data Fig. 2b–e). The resulting model comprises a single molecule of RTCB and PYROXD1, indicating that RTCB and PYROXD1 interact with a 1:1 stoichiometry (Fig. 1a,b). The interaction involves the C-terminal domain (CTD) of PYROXD1, whose function has hitherto remained unclear, and the active-site cleft of RTCB, spanning an extensive interface with a buried surface area of 1,715 Å[2].

The molecular reconstruction shows clear densities for the nicotinamide and flavin dinucleotides within the active center of PYROXD1 (Fig. 1c). Based on the kinetics of NADH oxidation and the dependence of RTCB–PYROXD1 interaction on NAD(P)H, we infer that the PYROXD1 is in the CTC state (that is, bound to $NAD^+–FADH^-$) when it interacts with RTCB. As in other flavoprotein oxidoreductases[22–27], the ligands are positioned by PYROXD1 such that the flavin ring of the $FADH^-$ molecule stacks with the nicotinamide ring of $NAD^+$, forming a π–π interaction (Fig. 1c). PYROXD1 coordinates $FADH^-$ with residues including Gly18, Ala21, Thr42, Ala43, Val87, Arg132, Trp239, Ser233, Ala234, Asn322, Asp358 and Trp376 (Fig. 1d and Extended Data Fig. 3a). In turn, $NAD^+$ interacts with residues Gly157, Ile158, Arg374, His410 and Thr412 (Fig. 1e and Extended Data Fig. 3b). The 2′-hydroxyl group of the adenosyl moiety of NADH is oriented towards the solvent and is not engaged in any protein contacts, suggesting that PYROXD1 can accommodate a phosphate group of NADPH without any steric hindrance, in agreement with previously measured enzyme kinetics data[19]. In sum, these structural and biochemical observations suggest that the interaction of PYROXD1 with RTCB is dependent on NAD(P)H binding in the PYROXD1 active-site center and the resulting formation of the CTC.

## C-terminal tail of PYROXD1 shields the RTCB catalytic center

The structure of the RTCB–PYROXD1 complex reveals that the PYROXD1 CTD engages with the catalytic cleft of RTCB, thereby blocking the substrate binding sites and rendering the catalytic center inaccessible (Fig. 2a). The interaction is centered on the C-terminal tail of PYROXD1 (residues Ile493–Asp500), which is unique to PYROXD1 and absent from other related NAD(P)H oxidases. The C-terminal tail is structurally disordered in free PYROXD1 but adopts an alpha-helical conformation upon interaction with RTCB, as it inserts into the catalytic center of RTCB and forms hydrophobic interactions with RTCB through Ile493, Ile495, Tyr498 and Phe499 of PYROXD1 (Fig. 2b). Furthermore, the C-terminal helix of PYROXD1 contacts RTCB by hydrogen bonding interactions of the C-terminal carboxylate group with the backbone amide group of Phe118 of RTCB (Fig. 2b), while the side chain of the C-terminal residue (Asp500 of PYROXD1) coordinates one of the two $Mg^{2+}$ ions (position B) in the RTCB catalytic center. This is consistent with the observation that the PYROXD1–RTCB interaction is dependent on the presence of divalent cations (Extended Data Fig. 1b), although the binding of divalent cations in the RTCB active site is also probably necessary to ensure overall electrostatic potential complementarity of the PYROXD1 and RTCB interaction surfaces.

To validate the structural determinants of the interaction, we generated mutant PYROXD1 proteins containing either individual alanine substitutions of the C-terminal residues Asp497–500Asp or C-terminal truncations (Δ500, Δ499-500, Δ498-500, Δ497-500 and Δ494-500) and performed in vitro affinity co-precipitation experiments using immobilized RTCB. Although the D500A, D497A and Δ500 mutant PYROXD1 proteins retained their ability to interact with RTCB at near-wild-type levels, mutations F499A and Y498A resulted in reduced co-precipitation by RTCB. Crucially, more extensive C-terminal truncations of PYROXD1 caused complete loss of RTCB binding (Fig. 2c). Altogether, these results reveal that the CTD of PYROXD1, and specifically the conserved C-terminal helix, is critical for the interaction with

**Table 1 | Cryo-EM data collection, refinement and validation statistics**

| | Human RTCB–PYROXD1 complex (EMD-17127),(PDB 8ORJ) |
|---|---|
| **Data collection and processing** | |
| Magnification | ×130,000 |
| Voltage (kV) | 300 |
| Electron exposure (e⁻/Å²) | Dataset 1, 64.592 |
| | Dataset 2, 63.163 |
| Defocus range (μm) | −1.0 to −2.4 (−0.2 steps) |
| Pixel size (Å) | 0.65 |
| Symmetry imposed | C1 |
| Initial particle images (no.) | 5,741,654 |
| Final particle images (no.) | 176,026 |
| Map resolution (Å) | 3.25 |
| FSC threshold | 0.143 |
| Map resolution range (Å) | 2.75–5.75 |
| **Refinement** | |
| Initial model used (PDB code) | 7P3B and 6ZK7 |
| Model resolution (Å) | 3.2 |
| FSC threshold | 0.143 |
| Model resolution range (Å) | 3.1–3.6 |
| Map sharpening B factor (Å²) | −145 |
| Model composition | |
| Non-hydrogen atoms | 6930 |
| Protein residues | 882 |
| Ligands | FDA 1 |
| | NAD 1 |
| | MG 2 |
| B factors (Å²) | |
| Protein | 30.00/132.60/79.91 |
| Ligand | 9.04/88.47/75.77 |
| R.m.s. deviations | |
| Bond lengths (Å) | 0.016 |
| Bond angles (°) | 1.818 |
| **Validation** | |
| MolProbity score | 1.74 |
| Clashscore | 5.39 |
| Poor rotamers (%) | 3.16 |
| Ramachandran plot | |
| Favored (%) | 97.70 |
| Allowed (%) | 2.30 |
| Disallowed (%) | 0.00 |

RTCB. Notably, deletion of Glu496 (ΔE496) has recently been shown to be associated with a connective tissue disorder and myopathy[21]. However, the ΔE496 mutant PYROXD1 protein retained its ability to interact with RTCB at near-wild-type levels in affinity co-precipitation experiments (Extended Data Fig. 1d), suggesting that the pathology is not caused by loss of RTCB protection.

Next, we tested the importance of the RTCB interaction interface in PYROXD1 for the protective function of PYROXD1 against oxidative inactivation of RTCB in the context of the tRNA-LC. To this end, we

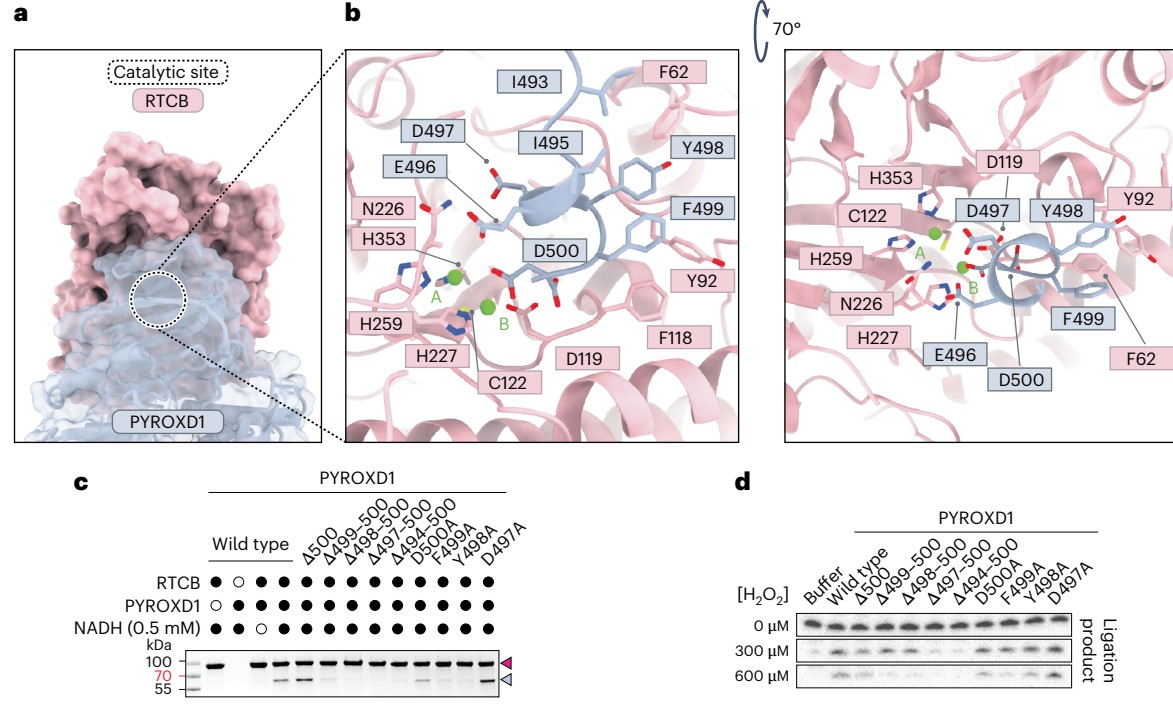

**Fig. 2 | The CTD of PYROXD1 occludes the catalytic center of RTCB. a**, Surface view of the RTCB–PYROXD1 interaction. The catalytic center of RTCB is indicated by the dashed circle. **b**, Detailed view of the interactions of the CTD of PYROXD1 within a 3.35 Å distance from the catalytic center of RTCB. **c**, In vitro pull-down experiment with PYROXD1 mutants and immobilized StrepII–GFP–RTCB.

Bound proteins were analyzed by SDS–PAGE and Coomassie Blue staining. The experiment was repeated three times with similar results. **d**, RNA ligase activity of tRNA-LC after treatment with $H_2O_2$ in the presence of wild-type or mutant PYROXD1 proteins. The experiment was repeated three times with similar results.

analyzed the ligase activity of tRNA-LC in an RNA circularization assay upon pre-incubating tRNA-LC with increasing concentrations of $H_2O_2$ in the presence of wild-type and mutant PYROXD1 proteins[19]. In the absence of hydrogen peroxide, tRNA-LC efficiently catalyzed RNA circularization in all reactions. By contrast, the complex only remained catalytically active upon $H_2O_2$ treatment in the presence of PYROXD1 protein variants capable of interacting with RTCB; ligase activity was not sustained by C-terminally truncated PYROXD1 proteins that did not interact with RTCB (Fig. 2d). Notably, the catalytic activity was reduced upon treatment with 600 µM hydrogen peroxide in the presence of the Δ500 or D500A PYROXD1 mutants compared to wild-type PYROXD1 (Fig. 2d), implicating the C-terminal residue of PYROXD1 in the protective activity. This is in agreement with the observed interactions of Asp500 of PYROXD1 with divalent cation B in the RTCB active site, which is in turn coordinated by the oxidation-sensitive active-site residue Cys122 in RTCB. Taken together, these results confirm that the interaction of the C-terminal tail of PYROXD1 with the catalytic center of RTCB is essential to protect RTCB from oxidative inactivation.

### Allosteric mechanism of RTCB–PYROXD1 interaction

Superposition of the RTCB–PYROXD1 complex structure with the crystal structure of NAD(P)H-free PYROXD1 (ref. 19) reveals structural rearrangements in PYROXD1 induced by NADH binding, and subsequent CTC formation and RTCB recruitment (Fig. 3a). In the free PYROXD1 crystal structure[19], a highly conserved loop within the FAD binding domain (residues Pro45–Arg76 of PYROXD1, hereafter referred to as loop 1), which contains an invariant sequence motif (motif 1), is well structured and positioned away from the FAD co-factor (Fig. 3b and Extended Data Fig. 4a). In the RTCB–PYROXD1 complex, the conformation of loop 1 is less well defined because of structural disordering; however, residual density in the cryo-EM map suggests that the loop shifts towards the PYROXD1 active center to avoid a steric clash with

RTCB (Extended Data Fig. 4b,c). The functional significance of loop 1 for RTCB binding could not be directly investigated because point mutations in motif 1 or deletion of the entire loop 1 render PYROXD1 insoluble, presumably by compromising its ability to bind FAD.

The structure of the RTCB–PYROXD1 complex further reveals that the NAD(P)H binding domain of PYROXD1 undergoes a structural rearrangement upon NADH binding and RTCB recruitment (Extended Data Fig. 4e). A long loop (residues Ala197–Glu251 of PYROXD1, hereafter referred to as loop 2), which connects the NAD(P)H binding and CTDs, is structurally disordered in the NAD(P)H-free crystal structure of PYROXD1 (ref. 19). However, in the RTCB–PYROXD1 complex, segments of loop 2 become partially ordered through interactions with the restructured NAD(P)H binding domain (Fig. 3b, right and Extended Data Fig. 4e–g)[19]. This positions a conserved motif (motif 2) within loop 2 (Extended Data Fig. 4d) such that the side chain of Trp239 of PYROXD1 contacts the flavin ring of FADH⁻ by side-on aromatic stacking (Fig. 3b and Extended Data Fig. 4f). In turn, Trp376 of PYROXD1, which caps the flavin ring of FAD in the crystal structure of free PYROXD1 and restricts its access to electron acceptors[19], is displaced away in the RTCB–PYROXD1 complex (Extended Data Fig. 4f). As this would otherwise lead to a steric clash with loop 1 in the absence of its restructuring, these observations suggest that NAD(P)H-dependent allosteric rearrangement of loop 2 induces, through Trp376 of PYROXD1, the disordering of loop 1, thereby exposing the RTCB interaction interface. Furthermore, the observed displacement of Trp376 upon allosteric restructuring of loop 2 also suggests that loop 2 modulates the NAD(P)H oxidase activity of PYROXD1 by regulating access to electron acceptors.

To validate the functional importance of loop 2, we generated mutant PYROXD1 proteins containing a point mutation in motif 2 (W239A), deletion of motif 2 (Δmotif 2) or a deletion of the entire loop 2 (Δloop 2) and measured their catalytic rates of NADH oxidation. Deletion of loop 2 in PYROXD1 resulted in the complete loss of

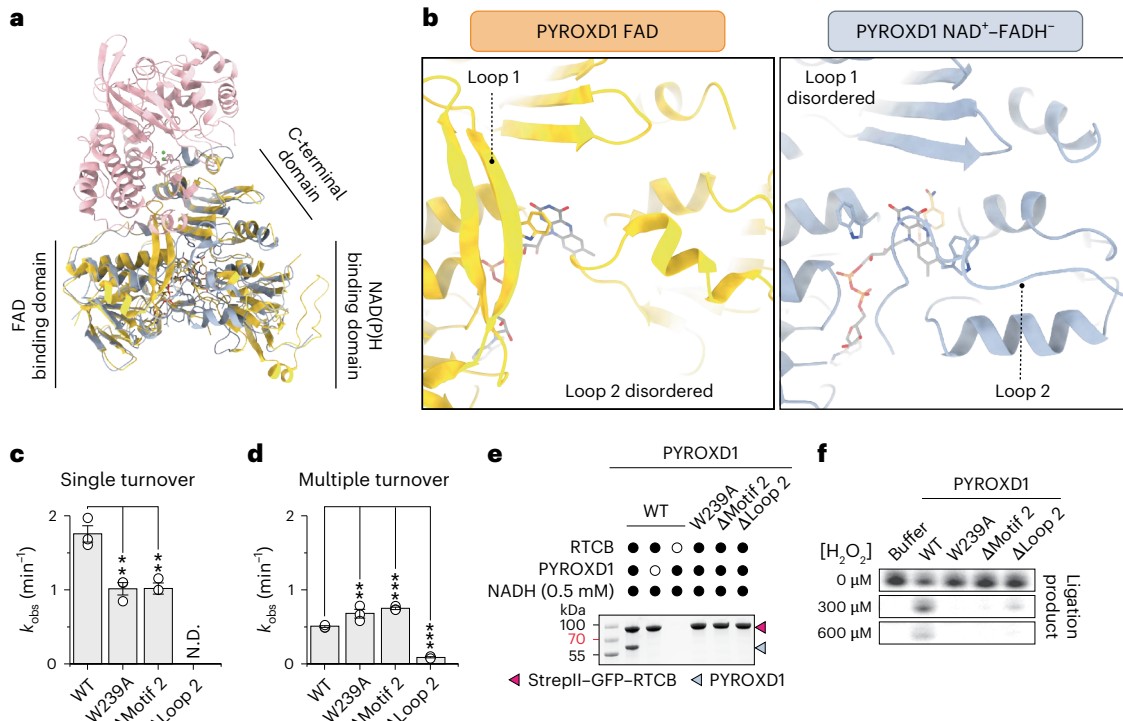

**Fig. 3 | Allosteric regulation of PYROXD1 mediates CTC formation and RTCB binding. a**, Structural superposition of PYROXD1 bound to FAD (orange, PDB 6ZK7)[19] and PYROXD1 (light blue) in complex with RTCB (pink). **b**, Detailed view of loop 1 and loop 2 regions of PYROXD1 bound to FAD (orange, PDB 6ZK7)[19] and PYROXD1 (light blue) in complex with RTCB. **c**, Single-turnover rates of NADH oxidation (monitored by change in absorbance at 340 nm) by PYROXD1 wild-type (WT) and loop 2 mutants under aerobic conditions. Data points represent the mean ± s.e.m. of three independent replicates. Significance was determined by using an unpaired two-tailed $t$-test. **$P ≤ 0.01$. Exact $P$ values are 0.0060 (W239A) and 0.0053 (Δmotif 2). **d**, Steady-state rates of NADH oxidation by PYROXD1

wild-type and loop 2 mutants under aerobic conditions. Data points represent the mean ± s.e.m. of three independent replicates. Significance was determined by using an unpaired two-tailed $t$-test. **$P ≤ 0.01$, ***$P ≤ 0.001$. Exact $P$ values are 0.0044 (W239A), <0.0001 (Δmotif 2) and <0.0001 (Δloop 2). **e**, In vitro pull-down experiment with PYROXD1 mutants and immobilized StrepII–GFP–RTCB. Bound proteins were analyzed by SDS–PAGE and Coomassie Blue staining. The experiment was repeated three times with similar results. **f**, RNA ligase activity of tRNA-LC after treatment with $H_2O_2$ in the presence of wild-type or mutant PYROXD1 proteins. The experiment was repeated three times with similar results.

oxidoreductase activity (Fig. 3c,d). Under single-turnover conditions, which reflected CTC formation, mutations of motif 2 (W239A and Δmotif 2) resulted in approximately twofold slower rates of NADH oxidation (Fig. 3c and Extended Data Fig. 5a). By contrast, under multiple turnover conditions, limited by the turnover rate of the CTC, mutations of motif 2 (W239A and Δmotif 2) resulted in ~1.5-fold faster rate of NADH oxidation (Fig. 3d and Extended Data Fig. 5b). Determination of multiple-turnover kinetics parameters revealed that the W293A mutation resulted in an approximately twofold increase in the Michaelis constant and an approximately threefold increase in the maximum velocity compared to wild-type PYROXD1 (Extended Data Fig. 5c). Together, these results implicate Trp293 of PYROXD1 in both NAD(P)H binding and its catalytic turnover.

Co-precipitation experiments in the presence of either NADH or NADPH revealed that mutations of motif 2 or deletion of loop 2 abolished interactions of PYROXD1 with RTCB (Fig. 3e and Extended Data Fig. 5d). In agreement with these results, the motif 2 and loop 2 PYROXD1 mutants failed to protect RTCB from oxidative inactivation in vitro (Fig. 3f). In sum, these data indicate that loop 2 allosterically modulates the oxidoreductase activity of PYROXD1, and hence its binding to RTCB, thereby underpinning its protective function against oxidative inactivation of RTCB.

## PYROXD1 protects RTCB before guanylylation by Archease

Superposition of the RTCB–PYROXD1 complex structure with the structure of *Pyrococcus horikoshii* RtcB bound to a 5′-hydroxyl DNA oligonucleotide[28] (Extended Data Fig. 6a,b) indicates that the PYROXD1–RTCB

interaction sterically hinders substrate RNA binding, thereby inhibiting the catalytic activity of RTCB. In turn, close inspection of a superposition of the RTCB–PYROXD1 complex with the crystal structure of human RTCB bound to GMP reveals steric clashes of PYROXD1 with a loop comprising residues Asp444–Glu474 of RTCB, which is structurally disordered in the PYROXD1–RTCB complex but adopts a defined conformation in the RTCB–GMP structure owing to interactions with bound GMP (Extended Data Fig. 6c). Additionally, binding of PYROXD1 to RTCB with non-covalently bound GMP, as well as guanylylated RTCB, would result in a steric clash between the side chain of Asp497 of PYROXD1 and the alpha-phosphate group of GMP (Extended Data Fig. 6d, e). Together, these structural observations suggest that PYROXD1 selectively interacts with the apo-enzyme form of RTCB, that is, before guanylylation of the active-site residue His428 of RTCB by Archease.

To test whether PYROXD1 interacts with the guanylylated or un-guanylylated forms of RTCB, we first pre-incubated RTCB in the presence of GTP, $Mn^{2+}$ ions and human Archease, generating a mixture of guanylylated and un-guanylylated RTCB. We subsequently performed a co-precipitation assay using immobilized PYROXD1 and analyzed the unbound and co-precipitated RTCB fractions by mass spectrometry. Guanylylated RTCB was substantially depleted from the co-precipitated (eluted) fraction compared to the unbound fraction and input fractions, suggesting that PYROXD1 preferentially associates with un-guanylylated, apo-RTCB (Fig. 4a and Extended Data Fig. 6f,g). As this implies that guanylylated RTCB cannot be protected by PYROXD1, we followed up on this observation and tested whether guanylylated RTCB is susceptible to oxidative inactivation. In this experiment,

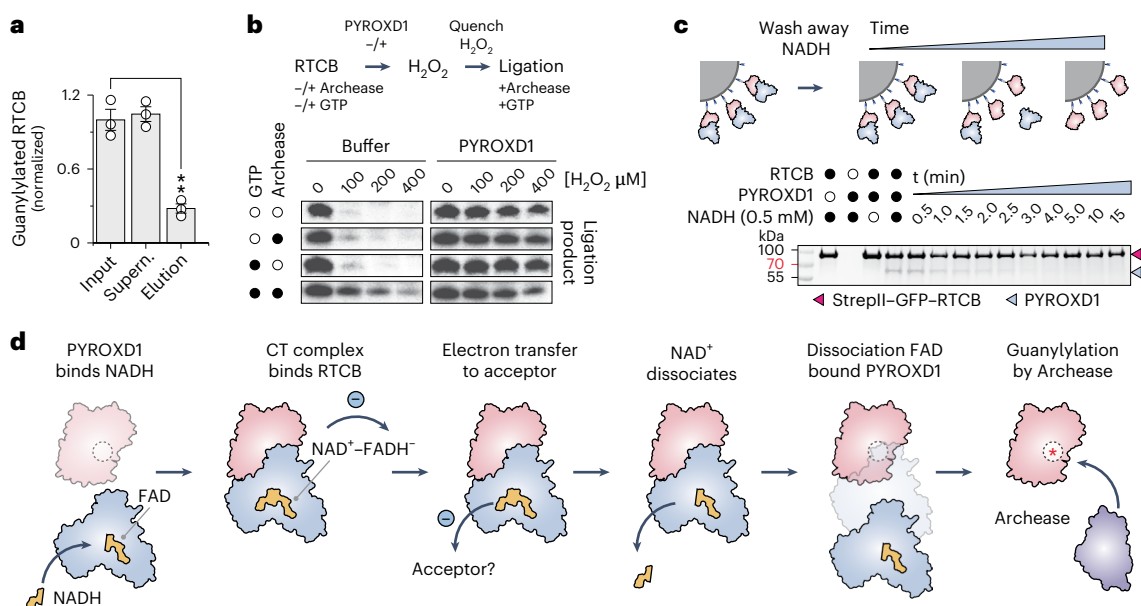

**Fig. 4 | Timed dissociation of the PYROXD1–RTCB complex enables Archease-mediated catalytic activation. a**, Normalized guanylylation levels of RTCB in the input, supernatant and elution fractions of a pull-down with GST–PYROXD1, as determined by LC–MS analysis. Data points represent the mean ± s.e.m. of three independent replicates. Significance was determined by using an unpaired two-tailed $t$-test. $**P \leq 0.01$. The exact $P$ value is 0.0015. **b**, In vitro RNA ligation assay to determine the effects of guanylylation on the oxidative inactivation of RTCB. The ligation assay was performed in a three-step procedure, in which tRNA-LC was pre-incubated with or without GTP and/or Archease (see labels on the left), followed by incubation with $H_2O_2$ in the presence or absence of PYROXD1. In the last step, all samples were supplemented with GTP and

Archease to allow for multiple-turnover ligation of the RNA substrate. Panels are a composite of separate experiments; see Extended Data Fig. 6h for individual gel panels. The experiment was repeated three times with similar results. **c**, In vitro pull-down experiment to monitor PYROXD1–RTCB complex dissociation. PYROXD1 was incubated with Strep-Tactin-immobilized StrepII–GFP–RTCB. Unbound PYROXD1 and NADH were washed out to initiate complex disassembly. At indicated times, bound proteins were analyzed by SDS–PAGE and Coomassie Blue staining. The experiment was repeated three times with similar results. **d**, Mechanistic model for the protection of RTCB by PYROXD1 against oxidative inactivation.

tRNA-LC samples containing apo-RTCB or pre-guanylylated RTCB were treated with $H_2O_2$ in the presence or absence of PYROXD1 and subsequently assayed for RNA ligase activity (Fig. 4b). In the absence of pre-guanylylation, RTCB was inactivated by hydrogen peroxide, which could be mitigated by the addition of PYROXD1 during peroxide treatment. By contrast, pre-guanylylated RTCB remained partially active upon $H_2O_2$ treatment in the absence of PYROXD1, suggesting that it is considerably less susceptible to peroxide-induced oxidative inactivation (Fig. 4b and Extended Data Fig. 6h). Together, these results suggest that PYROXD1 protects apo-RTCB, while guanylylated RTCB is intrinsically protected against oxidative inactivation.

#### PYROXD1 reoxidation controls RTCB release

The initial step of RTCB-catalyzed RNA ligation involves Archease-mediated guanylylation of the RTCB catalytic center. As Archease also interacts with the active site of RTCB, its binding to RTCB is mutually exclusive with PYROXD1, which necessitates the dissociation of the RTCB–PYROXD1 complex before RTCB guanylylation by Archease. In a co-precipitation experiment, Archease was not able to displace PYROXD1 from RTCB even when provided in a large molar excess together with GTP (Extended Data Fig. 6i). This result is consistent with previous observations that Archease interacts transiently with RTCB or tRNA-LC[9]. We reasoned that instead of direct competition with Archease, the dissociation of the RTCB–PYROXD1 complex requires reoxidation of the $NAD^+$–$FADH^-$ CTC within PYROXD1 by an electron acceptor, such as molecular oxygen, which results in FAD regeneration and $NAD^+$ release. To test this idea, we analyzed the dissociation kinetics of the RTCB–PYROXD1 complex in vitro under aerobic conditions. The complex was reconstituted in the presence of 0.5 mM NADH and immobilized on an affinity (Strep-Tactin) matrix via RTCB. Excess NADH was washed away and loss of PYROXD1 co-precipitation was monitored

over time (Fig. 4c). The estimated half-life of the RTCB–PYROXD1 complex upon NADH removal was ~2 min, in agreement with the measured rates of PYROXD1 CTC reoxidation (Figs. 3d and 4c). This suggests that PYROXD1 rapidly dissociates from RTCB upon reoxidation by an electron acceptor. Taken together, these data suggest a mechanism for the timed release of RTCB, enabling subsequent catalytic activation by Archease.

### Discussion

The RNA ligase activity of the tRNA-LC is essential for cellular RNA metabolism. The enzymatic mechanism of RTCB, the catalytic subunit of tRNA-LC, has anaerobic origins[16,17] and is dependent on a highly reactive cysteine residue, which makes the enzyme sensitive to inactivation under oxidative stress[18,19]. In metazoa, RTCB has co-evolved with the oxidoreductase PYROXD1, which protects RTCB from oxidative inactivation, whereas in land plants and fungi, RTCB has been functionally replaced by an evolutionarily unrelated RNA ligase with a distinct catalytic mechanism[19,29]. In this work, we combined single-particle cryo-EM with biochemical experiments to reveal the mechanistic basis for the protective function of PYROXD1. Our results further underscore the notion of co-evolution of PYROXD1 with RTCB as a mechanism to enable the tRNA-LC to function under aerobic conditions.

NAD(P)H was previously reported to directly affect the sensitivity of the tRNA-LC to oxidative inactivation by direct interaction with RTCB[19]. Our structural study reveals that within the RTCB–PYROXD1 complex, NAD(P)H instead occupies the active site of PYROXD1, thereby allosterically activating it to bind RTCB through an extensive molecular interface involving the C-terminal tail of PYROXD1 as well as divalent cations in the RTCB catalytic center. Based on our results, we propose a model in which PYROXD1 protects RTCB from oxidative inactivation by physically occluding its catalytic center (Fig. 4d)

and inhibiting metal ion exchange, thereby preventing oxidation of the invariant active-site cysteine residue (Cys122 in human RTCB) by excluding $Cu^{2+}$ ions and reactive oxygen species. PYROXD1 undergoes a conformational change upon binding of NAD(P)H, which facilitates the electron transfer from NAD(P)H to the FAD co-factor and results in the formation of the CTC. This conformational rearrangement positions two loops of PYROXD1 into the vicinity of the $NAD^+$–$FADH^-$ ligands within the protein, allowing for interaction with RTCB through elimination of steric clashes and exposure of the RTCB binding site and, finally, controlled turnover of the CTC. Consistent with this model, our biochemical data show that loop mutations that compromise the formation of the CTC or increase its turnover impair RTCB binding and the protective function of PYROXD1.

The catalytic activity of RTCB is dependent on the co-factor Archease, which catalyzes the initial guanylylation of a conserved histidine residue in the RTCB catalytic center. A recently determined structure of the RTCB–Archease complex reveals the mechanism of Archease-catalyzed guanylylation[10] and indicates that Archease and PYROXD1 interact with RTCB in a mutually exclusive manner. This is consistent with our observations that PYROXD1 selectively interacts with RTCB in the apo-enzyme form before guanylylation and that guanylylated RTCB is intrinsically protected from oxidative inactivation. Finally, our results suggest that the NAD(P)H oxidase activity of PYROXD1 serves as a molecular timer to provide controlled release of RTCB for the activation by Archease, thereby enabling the tRNA-LC to function under aerobic conditions[2,19]. As the coupling of tRNA-LC protection to the oxidoreductase activity of PYROXD1 also probably serves a regulatory function, this study establishes a paradigm for redox-dependent control of RNA processing pathways.

## Online content

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

## Methods

### Plasmid DNA constructs and site-specific mutants

DNA fragments encoding human Archease (UniProt Q8IWT0), human PYROXD1 (UniProt Q8WU10) and human RTCB (UniProt Q9Y3I0) were obtained as described previously[9,19]. In brief, human PYROXD1 was amplified from HeLa cDNA by PCR. The amplified sequence was ligated into the pGEX-6P-1 vector (GE Healthcare) upon cleavage with the restriction enzymes BamHI and XhoI. The resulting construct carries an amino-terminal glutathione tag followed by a 3C Precision protease cleavage site. Mutants of PYROXD1 were generated by Gibson assembly[30], using gBlocks synthesized by IDT. The DNA sequence encoding human Archease, spanning residues 27–168, was inserted into the 2M-T (Addgene, 29708) plasmid using ligation-independent cloning, resulting in a construct carrying an N-terminal hexahistidine tag followed by a maltose binding protein (MBP) and a tobacco etch virus (TEV) protease cleavage site. The DNA sequence encoding human RTCB was inserted into the 438B (Addgene, 30115) plasmid and the 438-Rgfp (Addgene, 55221) plasmid, using ligation-independent cloning. The resulting constructs carry an N-terminal hexahistidine tag followed by a TEV protease cleavage site or an N-terminal StrepII tag followed by a super folder green fluorescent protein and a TEV protease cleavage site, respectively.

### Expression and purification of PYROXD1

For expression and purification of PYROXD1, *Escherichia coli* BL21 (DE3) (Invitrogen, C601003) cells were transformed with pGEX-6P-1 plasmid encoding for GST–PYROXD1 wild-type and mutant variants. Bacterial cells were grown at 37 °C, 130 rpm in LB medium supplemented with 100 μg ml$^{-1}$ ampicillin. When cultures reached an optical density of 0.6 at 600 nm ($OD_{600}$), the temperature was decreased to 18 °C, and protein expression was induced with 0.1 mM isopropyl-b-D-thiogalactosidase, after which cell growth was continued for another 18 h. Bacterial cells were collected by centrifugation and flash-frozen in liquid nitrogen for storage at −80 °C.

Cell pellets were thawed and resuspended in lysis buffer containing 50 mM Tris-HCl pH 8.0, 150 mM NaCl, 5 mM MgCl$_2$ supplemented with 1 mM AEBSF, 0.1% (v/v) Tween-20, 1 mg ml$^{-1}$ chicken egg white lysozyme and one proteinase inhibitor tablet per 100 ml (cOmplete Protease Inhibitor, EDTA free). After 30 min of incubation at 4 °C with gentle shaking, cells were lysed by ultrasonication, and the lysate was cleared by centrifugation at 40,000g for 30 min at 4 °C. The cleared lysate was incubated with Glutathione Sepharose beads (Cytiva, GE17-0756-01), equilibrated with lysis buffer, for 2 h at 4 °C with gentle rocking. Next, the beads were washed five times with lysis buffer and resuspended in one column volume of lysis buffer.

To elute PYROXD1 from the beads, 1 mg of GST-3C Precision protease was added and incubated for 18 h at 4 °C. Subsequently, the beads were loaded onto a gravity flow column and washed with three column volumes of lysis buffer. The collected flow-through was pooled and concentrated using 30 kDa molecular weight cut-off centrifugal filters (Merck Millipore). The concentrated sample was loaded onto a HiLoad Superdex S75 (16/600) column (Cytiva, 28989333) equilibrated with a buffer containing 20 mM HEPES pH 8.0, 150 mM NaCl and 1 mM TCEP, yielding pure and monodisperse protein. Peak fractions were pooled, concentrated using 30 kDa molecular weight cut-off centrifugal filters (Merck Millipore) and flash-frozen in liquid nitrogen for storage at −80 °C.

For the purification of GST–PYROXD1, the protocol described above was altered as follows. After incubating the cleared lysate with Glutathione Sepharose beads, the beads were transferred to a gravity flow column and washed five times with lysis buffer. Proteins were eluted from the beads in five fractions by the addition of one column volume of lysis buffer supplemented with 10 mM GSH. Eluted fractions were analyzed by SDS–PAGE, and GST–PYROXD1-containing fractions were pooled, after which proteins were dialyzed for 18 h at 4 °C against

lysis buffer. The next day, the proteins were collected and concentrated using 30 kDa molecular weight cut-off centrifugal filters (Merck Millipore) for size-exclusion chromatography.

### Expression and purification of Archease

For expression and purification of Archease, *E. coli* BL21(DE3) Rosetta2 (Sigma-Aldrich, 71402) cells were transformed with the 2M-T plasmid encoding for His$_6$–MBP–Archease. Cell cultures were grown at 37 °C, 130 rpm in LB medium supplemented with 100 μg ml$^{-1}$ ampicillin. When cultures reached an $OD_{600}$ of 0.6, the temperature was decreased to 18 °C and protein expression was induced with 0.2 mM isopropyl-b-D-thiogalactosidase, after which cell growth was continued for another 18 h. Bacterial cells were collected by centrifugation and flash-frozen in liquid nitrogen for storage at −80 °C.

Cell pellets were thawed and resuspended in lysis buffer containing 20 mM Tris pH 8.0, 500 mM NaCl, 5 mM imidazole, 1 μg ml$^{-1}$ pepstatin and 200 μg ml$^{-1}$ AEBSF. After 30 min of incubation at 4 °C with gentle shaking, cells were lysed by ultrasonication and the lysate was cleared by centrifugation at 40,000g for 30 min at 4 °C. The cleared lysate was loaded onto two 5 ml Ni-NTA Superflow cartridges (QIAGEN, 30761) equilibrated with lysis buffer and washed with ten column volumes of washing buffer containing 20 mM Tris pH 8.0, 500 mM NaCl and 10 mM imidazole. Next, proteins were eluted with elution buffer containing 20 mM Tris pH 8.0, 500 mM NaCl and 250 mM imidazole. Protein-containing fractions were pooled and dialyzed for 18 h at 4 °C against 20 mM Tris pH 8.0, 250 mM KCl and 5% glycerol in the presence of His$_6$–TEV protease.

To remove the affinity tag, remaining impurities, uncleaved protein and His$_6$–TEV protease, the sample was re-applied onto the Ni-NTA Superflow cartridges and the flow-through was pooled and concentrated using 10 kDa molecular weight cut-off centrifugal filters (Merck Millipore). The concentrated sample was loaded onto a HiLoad Superdex S75 (26/600) column (Cytiva, 28989334) equilibrated with a buffer containing 20 mM HEPES pH 8.0, 250 mM KCl and 1 mM dithiothreitol (DTT), yielding pure and monodisperse protein. Peak fractions were pooled, concentrated using 10 kDa molecular weight cut-off centrifugal filters (Merck Millipore) and flash-frozen in liquid nitrogen for storage at −80 °C.

### Expression and purification of RTCB and StrepII–GFP–RTCB

For expression and purification of RTCB and StrepII–GFP–RTCB, recombinant baculoviruses encoding for His$_6$–MBP–RTCB or StrepII–GFP–RTCB, respectively, were generated using the Bac-to-Bac Baculovirus expression system (Thermo Scientific). These Baculoviruses were used to infect Sf9 insect cells (Thermo Scientific, 11496015) at a density of $1.0 × 10^6$ ml$^{-1}$. After infection, cells were cultured for 60 h at 27 °C, 90 rpm in SF-4 Baculo Express insect cell culture medium (BioConcept, 9-00F38-N). Insect cells were collected by centrifugation and flash-frozen in liquid nitrogen for storage at −80 °C.

To purify RTCB, cell pellets were thawed and resuspended in lysis buffer containing 20 mM Tris pH 8.0, 150 mM NaCl, 5 mM imidazole supplemented with 0.1% Tween-20 and one proteinase inhibitor tablet per 50 ml (cOmplete Protease Inhibitor, EDTA free). Cells were lysed by ultrasonication, and the lysate was cleared by centrifugation at 40,000g for 30 min at 4 °C. The cleared lysate was loaded onto Ni-NTA Superflow resin (QIAGEN, 30410) equilibrated with lysis buffer and washed with ten column volumes of washing buffer containing 20 mM Tris pH 8.0, 500 mM NaCl and 10 mM imidazole. Next, proteins were eluted with elution buffer containing 20 mM Tris pH 8.0, 500 mM NaCl and 250 mM imidazole. Protein-containing fractions were pooled and dialyzed for 18 h at 4 °C against 20 mM HEPES pH 8.0, 500 mM KCl and 3 mM MgCl$_2$ in the presence of His$_6$–TEV protease.

To remove the affinity tag, remaining impurities, uncleaved protein and His$_6$–TEV protease, the sample was re-applied onto Ni-NTA Superflow resin and the flow-through was pooled and concentrated

using 10 kDa molecular weight cut-off centrifugal filters (Merck Millipore). The concentrated sample was loaded onto a HiLoad Superdex S200 (16/600) column (Cytiva, GE28-9893-35) equilibrated with a buffer containing 20 mM HEPES pH 8.0, 250 mM KCl and 1 mM DTT, yielding pure and monodisperse protein. Peak fractions were pooled, concentrated using 10 kDa molecular weight cut-off centrifugal filters (Merck Millipore) and flash-frozen in liquid nitrogen for storage at −80 °C.

To purify StrepII–GFP–RTCB, cell pellets were thawed and resuspended in lysis buffer containing 20 mM Tris pH 8.0, 150 mM NaCl, supplemented with 0.1% Tween-20 and one proteinase inhibitor tablet per 50 ml (cOmplete Protease Inhibitor, EDTA free). Cells were lysed by ultrasonication and the lysate was cleared by centrifugation at 40,000$g$ for 30 min at 4 °C. The clarified lysate was applied to a gravity column containing 5 ml Strep-Tactin resin (IBA, 2-1201-002) and washed with ten column volumes of wash buffer containing 20 mM HEPES pH 8.0, 500 mM KCl, 1 mM DTT and 2 mM MgCl$_2$. Proteins were eluted with wash buffer supplemented with 2.5 mM desthiobiotin. Eluted proteins were pooled and concentrated using 10 kDa molecular weight cut-off centrifugal filters (Merck Millipore). The concentrated sample was loaded onto a HiLoad Superdex S200 (16/600) column (Cytiva, GE28-9893-35) equilibrated with a buffer containing 20 mM HEPES pH 8.0, 500 mM KCl, 2 mM MgCl$_2$ and 1 mM DTT, yielding pure and monodisperse protein. Peak fractions were pooled, concentrated using 10 kDa molecular weight cut-off centrifugal filters (Merck Millipore) and flash-frozen in liquid nitrogen for storage at −80 °C.

## RTCB–PYROXD1 complex sample preparation and cryo-EM data collection

Before grid preparation for cryo-EM, thawed protein samples of RTCB and PYROXD1 were incubated at a 1:1 molar ratio by mixing 100 μM RTCB with 100 μM PYROXD1 in 100 μl buffer containing 20 mM HEPES pH 8.0, 150 mM NaCl, 0.5 mM TCEP, 5 mM MgCl$_2$ and 0.5 mM NADH. After 30 min of incubation on ice, samples were loaded onto a Superdex 200 (10/300) size-exclusion chromatography column (GE Healthcare) equilibrated with a buffer containing 20 mM HEPES pH 8.0, 150 mM NaCl, 0.5 mM TCEP, 5 mM MgCl$_2$ and 0.5 mM NADH (Sigma-Aldrich, N8129). Protein complex-containing fractions were collected and concentrated using 10 kDa molecular weight cut-off centrifugal filters (Merck Millipore). The concentrated protein was split into two samples: a sample at a protein concentration of 1.5 mg ml$^{-1}$ that was supplemented with 0.01% octyl-beta-glucoside (dataset 1) and a sample at a protein concentration of 0.6 mg ml$^{-1}$ (dataset 2). To each 200-mesh holey carbon grid (Au R1.2/1.3, Quantifoil Micro Tools), 2.5 μl of sample was applied and blotted for 4 s at 80% humidity and 4 °C. Grids were plunge-frozen in liquid ethane using a Vitrobot Mark IV plunger (Thermo Scientific) and stored in liquid nitrogen until cryo-EM data collection. Cryo-EM data collection was performed on a FEI Titan Krios microscope equipped with a Gatan K3 direct electron detector (University of Zurich) operated at 300 kV in super-resolution counting mode and controlled through Gatan Digital micrograph (v.1.84.1282). Data acquisition was performed using the EPU Automated Data Acquisition Software for Single Particle Analysis from ThermoFisher (v.2.9.0.1519REL) with three shots per hole at a defocus range of −1.0 μm to −2.4 μm (0.2 μm steps). The final datasets comprised a total of 14,516 micrographs (dataset 1, 5,114 micrographs; dataset 2, 9,302 micrographs) at a calibrated magnification of ×130,000 and a super-resolution pixel size of 0.325 Å. Micrographs for dataset 1 were exposed for 1.01 s with a total dose of 64.592 e$^-$ Å$^{-2}$ over 38 subframes, whereas the micrographs for dataset 2 were exposed for 1.01 s with a total dose of 66.459 e$^-$ Å$^{-2}$ over 38 subframes.

## cryo-EM data processing and model building

Cryo-EM data was processed using cryoSPARC (v.3.3.2)[31]. A total of 14,516 micrographs were imported and motion-corrected with patch motion correction (multi), after which the contrast transfer function (CTF) values of the micrographs were estimated using patch CTF estimation (multi). Micrographs were curated and micrographs with a resolution estimate >5 Å and a defocus >2.4 μm were discarded from the dataset (742 micrographs), yielding 13,774 micrographs for further processing steps. Next, an initial set of particles was picked with Blob Picker using an elliptical blob and a minimum and maximum particle diameter of 50 Å and 150 Å, respectively. After extraction of the particles with a box size of 288 × 288 pixels, particles were subjected to 2D classification to generate templates for picking (five templates). After template-based picking with a particle diameter of 100 Å, particles were extracted and subjected to 2D classification with a circular mask of 110 Å. All particles were used to generate five ab initio volumes, which were used for supervised 3D classification with five classes. Classes were inspected visually using UCSF Chimera[32], and the particles of the best class were used to generate four ab initio volumes. The particles and corresponding volume of the best class were subjected to unsupervised 3D classification with four classes, which were inspected visually using UCSF Chimera[32]. The best class was used for variability analysis with three modes, filtered at a resolution of 5 Å. After visual inspection of the variability analysis using UCSF Chimera[32], two volumes were used for heterogeneous refinement with two classes. Classes were inspected, and the particles and volume of the best class were subjected to non-uniform refinement with optimization of CTF parameters enabled. The final map was sharpened with a $B$ factor of −145. The local resolution was estimated based on the resulting map using the local resolution function of cryoSPARC and displayed on the map using UCSF Chimera[32]. The structural model of the RTCB–PYROXD1 was built in Coot (v.0.9.2)[33], using crystallographic structures of RTCB[9] and PYROXD1 (ref. [19]) as starting models. The model was refined over multiple rounds using Phenix[34,35] with global minimization, atomic displacement parameter refinement, reference model restraints and secondary structure restraints enabled. The quality of the atomic model, including protein geometry, Ramachandran plots, clash analysis and model cross-validation, was assessed with MolProbity and the other validation tools in Phenix[35,36]. The refinement statistics of the final model are listed in Table 1. Figures of maps, models and the calculations of map contour levels were generated using UCSF ChimeraX[37].

## In vitro pull-down assays of RTCB–PYROXD1 complexes

To immobilize RTCB, 0.2 nmol of StrepII–GFP–RTCB was diluted in 200 μl of binding buffer containing 20 mM HEPES pH 8.0, 150 mM KCl, 0.1% Tween-20, 1 mM TCEP, 2.5 mM MgCl$_2$, 2.5 mM MnCl$_2$ and 0.5 mM NADH (Sigma-Aldrich, N8129). This solution was incubated by gently rocking for 30 min at 4 °C with 2 μl (packed volume) of MagStrep 'type 3' XT beads (IBA, 2-4090-002). The supernatant was removed and the beads were washed twice with 500 μl of binding buffer. Next, 0.4 nmol of PYROXD1 in 200 μl of reaction buffer containing HEPES pH 8.0, 150 mM KCl, 0.1% Tween-20, 1 mM TCEP and 0.5 mM NADH (Sigma-Aldrich, N8129) supplemented with 2.5 mM MgCl$_2$, and 2.5 mM MnCl$_2$ was added to the washed beads and incubated gently by rocking for 30 min at 4 °C. To test the influence of metals on PYROXD1 binding, the reaction buffer was supplemented with either 5 mM (MgCl$_2$, MnCl$_2$, NiCl$_2$, CoCl$_2$, CaCl2) or 0.1 mM (ZnCl$_2$, CuCl$_2$) of divalent metal. After incubation, the beads were washed three times with 500 μl of reaction buffer. After washing, the beads were resuspended in 40 μl of 1× SDS loading dye (45 mM Tris pH 6.8, 10% glycerol, 1% SDS, 50 mM DTT, 0.002% Bromophenol Blue). The samples were incubated at 95 °C for 5 min and 10 μl of the sample was resolved on a 4–15% Mini-PROTEAN TGX Precast Gel (Biorad) and stained with Coomassie Blue.

## RNA ligation assays in the presence of PYROXD1 variants

Ligation assays were carried out following a two-step procedure using purified recombinant tRNA-LC, prepared as described previously[9]. In the first step, the oxidation step, 29 nM tRNA-LC (containing RTCB,

DDX1, CGI-99 and FAM98b) was incubated with 290 nM Archease, 100 nM PYROXD1 and $H_2O_2$ at various concentrations for 1 h at 20 °C at 600 rpm in a reaction buffer containing 30 mM HEPES-KOH (pH 7.4), 100 mM KCl, 5 mM $MgCl_2$, 0.1 mM AEBSF, 10% glycerol and 1% NP-40, which was supplemented with 440 µM NADPH. Separately, a solution containing 12.5 mM TCEP, 100 mM KCl, 2.9 mM $MgCl_2$, 7.5 mM ATP, 0.5 mM GTP, 250 µM $ZnCl_2$, RNasin Ribonuclease Inhibitors (1 ml per 500 ml) and 65% (v/v) glycerol was mixed (2/1, v/v) with the radiolabelled oligoribonucleotide to generate a reaction cocktail. The reaction cocktail was then mixed with the samples from the oxidation step in a 3:2 (v/v) ratio, respectively. In the second step of the procedure, the ligation step, samples were incubated for 45 min at 30 °C. The final concentrations of the critical components in the ligation step were 12 nM tRNA-LC, 120 nM Archease, 3.2 mM ATP, 230 µM GTP, 180 µM NADPH and 40 nM PYROXD1. The ligation reactions were then quenched with 2x formamide solution (1:1, v/v), composed of 90% formamide, 50 mM EDTA, 1 ng ml$^{-1}$ Bromophenol Blue, 1 ng ml$^{-1}$ xylene cyanol and incubated for 1 min at 95 °C. The reactions were resolved on a 15% denaturing polyacrylamide gel and detected by autoradiography using a Typhoon 5 phosphorimager.

The levels of oxidative inactivation of the tRNA-LC are dependent on the ratio of NAD(P)H to NAD(P)$^+$, the presence of trace metal ions and the concentration of reducing agents in the reaction buffer. To obtain reproducible results, NAD(P)H stocks must be freshly prepared before the experiment from NAD(P)H that has not been stored long-term. Additionally, the assay is sensitive to the levels of trace metals that are present in the water; hence, buffers need to be prepared in bulk, aliquoted and stored at −20 °C until use. As tRNA-LC is prepared in a buffer with a reducing agent, it is important to maintain a constant dilution factor between experiments to account for the presence of residual reducing agent in the solution. Lastly, fresh glycerol should be used for the assays and storage, as glycerol that has been exposed to light and high temperatures may contain traces of electrophilic dehydration or oxidation products that inactivate tRNA-LC.

### In vitro competition assays

For competition assays, 0.2 nmol of StrepII–GFP–RTCB was diluted in 200 µl of binding buffer. This solution was incubated by gently rocking for 30 min at 4 °C with 2 µl (packed volume) of MagStrep 'type 3' XT beads (IBA, 2-4090-002). The supernatant was removed and the beads were washed twice with 500 µl of binding buffer. Next, 0.4 nmol of PYROXD1 and 2 nmol Archease in 200 µl of reaction buffer containing HEPES pH 8.0, 150 mM KCl, 0.1% Tween-20, 1 mM TCEP, 2.5 mM $MgCl_2$ and 2.5 mM $MnCl_2$ supplemented with 0.5 mM NADH (Sigma-Aldrich, N8129), 1 mM GTP (Roth, K056.4) or 1 mM GMP (Jena Biosciences, JBS-NU-1028) was added to the washed beads and incubated gently by rocking for 30 min at 4 °C. After incubation, the beads were washed three times with 500 µl of reaction buffer. After washing, the beads were resuspended in 40 µl of 1× SDS loading dye (45 mM Tris pH 6.8, 10% glycerol, 1% SDS, 50 mM DTT and 0.002% Bromophenol Blue). The samples were incubated at 95 °C for 5 min and 10 µl of the sample was resolved on a 4–15% Mini-PROTEAN TGX Precast Gel (Biorad) and stained with Coomassie Blue dye.

### Spectroscopic assays

For spectroscopic analysis of NADH oxidation in single-turnover conditions, PYROXD1 was diluted to 25 µM in 100 µl of buffer containing 20 mM Tris-HCl pH 8.0, 150 mM KCl and 0.05% Tween-20. Diluted samples were transferred to a 96-well plate (Corning, 3635) and placed into a PHERAstar FSX plate reader (BMG Labtech). Next, NADH (Sigma-Aldrich, N8129) was injected into the wells to a final concentration of 25 µM, and absorbance at 340 nm was measured over 30 min at 3 min intervals at room temperature (22 °C). For control experiments, the absorbance at 340 nm of 25 µM NADH (Sigma-Aldrich, N8129), 25 µM FAD (Sigma-Aldrich, F6625) and a mixture of 25 µM

NADH and 25 µM FAD were tracked over 30 min at 3 min intervals at room temperature. For spectroscopic analysis of NADH oxidation under multiple turnover conditions, PYROXD1 was diluted to 2.5 µM in 100 µl of buffer containing 20 mM Tris-HCl pH 8.0, 150 mM KCl and 0.05% Tween-20. Diluted samples were transferred to a 96-well plate (Corning, 3635) and placed into a PHERAstar FSX plate reader (BMG Labtech). Next, NADH (Sigma-Aldrich, N8129) was injected into the wells to a final concentration of 250 µM, and absorbance at 340 nm was measured over 30 min at 3 min intervals at room temperature. Data were plotted using Origin Pro (2018) and statistical analysis was performed in Prism GraphPad (v.9).

### Co-precipitation of guanylylated RTCB

For the guanylylation reaction, 4 nmol of RTCB was incubated with 20 nmol Archease in 200 µl guanylylation buffer containing 20 mM HEPES pH 8.0, 100 mM NaCl, 5 mM DTT, 2.5 mM $MnCl_2$ and 0.5 mM GTP (Roth, K056.4) for 60 min at 25 °C. After the incubation, 25 µl of the sample was removed, flash-frozen in liquid nitrogen and stored at −80 °C until subsequent analysis by liquid-chromatography–mass spectrometry (LC–MS). In parallel, 25 µl (packed volume) of Glutathione Sepharose 4 Fast Flow beads (Sigma-Aldrich, GE17-5132-01) was equilibrated in guanylylation buffer, after which the beads were incubated with 1 nmol GST–PYROXD1 in 200 µl of guanylylation buffer for 30 min at 4 °C, gently rocking. Subsequently, the beads were washed two times with 200 µl guanylylation buffer and added to the guanylylation sample. After supplementing the mixture with 0.5 mM NADH (Sigma-Aldrich, N8129), the sample was incubated for 15 min at 4 °C, gently rocking. After the incubation, the sample was centrifuged for 2 min at 500g and 4 °C, and the supernatant was recovered, flash-frozen in liquid nitrogen and stored at −80 °C for subsequent analysis by LC–MS. Next, the beads were washed three times with 500 µl guanylylation buffer supplemented with 0.5 mM NADH, followed by elution of the proteins with 40 µl of elution buffer containing 20 mM HEPES pH 8.0, 150 mM KCl, 10 mM GSH and 2.5 mM $MnCl_2$. The eluted proteins were flash-frozen in liquid nitrogen and stored at −80 °C for subsequent analysis by LC–MS. The experiment was carried out in triplicate.

### Quantification of guanylylated RTCB by LC–MS

To quantify the levels of guanylylated RTCB by LC–MS, samples were diluted 1:3 with water, after which 5 µl was injected into an ACQUITY UPLC BioResolve-RP-mAb polyphenyl column (450 Å, 2.7 µm, 2.1 mm × 150 mm) column (Waters). To separate proteins, a gradient was applied from 0.1% difluoroacetic acid in water to 0.1% difluoroacetic acid in acetonitril/75% 2-propanol at 60 min over 15 min. LC–MS analysis was performed on a SYNAPT G2-Si mass spectrometer coupled to an ACQUITY UPLC station. Data was plotted using Origin Pro (2018) and statistical analysis was performed in Prism GraphPad (v.9).

### Ligation assays to assess the effects of guanylylation on oxidative inactivation of RTCB

These experiments were carried out using a three-step procedure. In the first step (guanylylation), 98 nM of recombinant tRNA-LC (containing RTCB, DDX1, CGI-99 and FAM98b) was incubated for 1 h at 30 °C, 600 rpm with 980 µM ATP and, where designated, 980 nM Archease and/or 220 µM GTP in a reaction buffer containing 30 mM HEPES-KOH (pH 7.4), 100 mM KCl, 5 mM $MgCl_2$, 0.1 mM AEBSF, 10% glycerol and 1% NP-40. Next, in the second step (oxidation), NADPH, $H_2O_2$ and PYROXD1 were added, resulting in the following concentrations of the critical components: 29 nM tRNA-LC, 290 nM Archease, 290 µM ATP, 67 µM GTP, 440 µM NADPH, 100 nM PYROXD1 and the designated concentrations of $H_2O_2$. Subsequently, solutions of GTP or Archease were added to supplement the component missing in the guanylylation and oxidation steps to ensure that all reactions enter the final ligation step with the same reagent compositions. The ligation step was carried out and the products were analyzed as described above.

## Dissociation kinetics analysis of RTCB–PYROXD1

For the RTCB dissociation assay, 2 nmol 2× StrepII–GFP–RTCB and 4 nmol PYROXD1 were diluted in 3 ml binding buffer containing 150 mM KCl, 20 mM HEPES-KOH pH 8, 1 mM TCEP, 0.1% Tween-20, 5 mM $MgCl_2$ and 0.5 mM NADH and then incubated for 10 min on ice. Next, 400 µl slurry of Magnetic Strep-Tactin beads (MagStrep 'type 3' XT beads, IBA 2-4090-002) equilibrated with binding buffer and added to the 2× StrepII–GFP–RTCB and PYROXD1 mixture followed by incubation at 1 h at 4 °C, rocking gently. Subsequently, the supernatant was removed and the magnetic beads were transferred to a 1.5 ml reaction tube. After washing the beads twice with 1 ml wash buffer containing 150 mM KCl, 20 mM HEPES-KOH pH 8, 1 mM TCEP, 0.1% Tween-20 and 5 mM $MgCl_2$, the resin was equally distributed to multiple 1.5 ml reaction tubes and the resin resuspended in 400 µl wash buffer. At the indicated time points, the supernatant was removed, the resin was washed twice with 400 µl wash buffer and the beads were resuspended in 40 µl SDS sample buffer.

## Reporting summary

Further information on research design is available in the Nature Portfolio Reporting Summary linked to this article.

## Data availability

The proteins used in this study are associated with the following UniProt accession codes: human Archease (Q8IWT0), human PYROXD1 (Q8WU10) and human RTCB (Q9Y3I0). Atomic coordinates and cryo-EM maps for the human RTCB–PYROXD1 complex (PDB 8ORJ, EMD-17127) have been deposited in the PDB and EMDB databases and are publicly available as of the date of publication. Source data are provided with this paper.

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

## Acknowledgements

We thank M. Sawicka, S. Sorrentino and the University of Zurich (UZH) Center for Microscopy and Image Analysis for technical support with cryo-EM data acquisition. We thank J. Peschek and J. Gerber for fruitful discussions and sharing unpublished data. We thank S. Gradia (UC Berkeley MacroLab) for providing ligation-independent cloning vectors. We thank S. Chesnov from the functional genomics center in Zurich for assistance with processing and data acquisition of RTCB samples for guanylylation analysis. L.L. was funded by a Horizon 2020 Marie Skłodowska-Curie Individual Fellowship (project no. 845268, MSOPGDM). A.K. was funded by the Boehringer Ingelheim Fonds PhD Fellowship and the Forschungskredit program of the University of Zurich (grant no. FK-18-033). The work in the Martinez group was funded by the Medical University of Vienna and the Austrian Science Fund ('Fonds zur Förderung der wissenschaftlichen Forschung' FWF) Stand-Alone Projects P29888 and P34895. M.J. is an International Research Scholar of the Howard Hughes Medical Institute, Vallee Scholar of the Vallee Foundation and member of the Swiss National Competence Center for Research 'RNA & Disease'.

## Author contributions

A.K., I.A., J.M., L.L. and M.J. conceived the study. A.K., F.A., F.B., I.A. and L.L. purified the proteins. L.L. collected and processed the cryo-EM data. A.F., A.L., A.K., F.A., F.B., I.A., L.L., L.R. and M.P. performed the biochemical experiments. L.L. and M.J. wrote the paper with input from the remaining authors.

## Funding

## Competing interests

The authors declare no competing interests.

## Additional information

**Extended data** is available for this paper at https://doi.org/10.1038/s41594-025-01516-6.

**Correspondence and requests for materials** should be addressed to Martin Jinek.

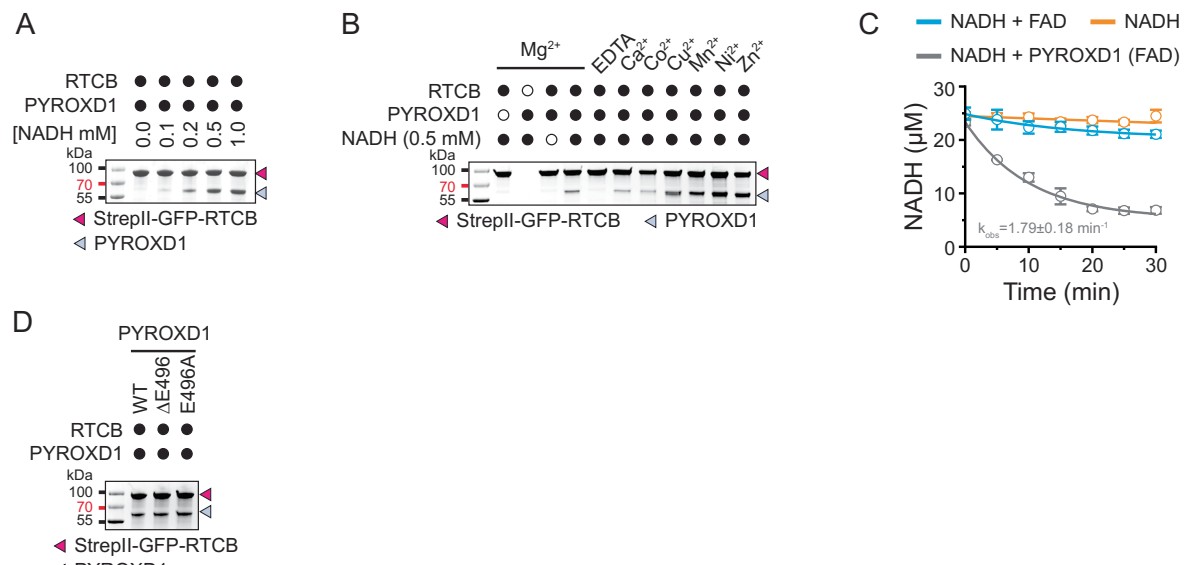

**Extended Data Fig. 1 | RTCB-PYROXD1 complex formation and spectroscopic analysis of NADH turnover by PYROXD1. (a)** Co-precipitation of PYROXD1 by immobilized StrepII-GFP-RTCB in the presence of increasing concentrations of NADH. Strep-Tactin beads were washed to remove unbound PYROXD1, and bound proteins were analyzed by SDS-PAGE and Coomassie blue staining. This experiment was repeated three times with similar results. **(b)** In vitro pull-down experiment with PYROXD1 mutants and StrepII-GFP-RTCB in the presence of divalent metal ions. Strep-Tactin beads were washed to remove unbound PYROXD1, and bound proteins were analyzed by SDS-PAGE and Coomassie blue staining. This experiment was repeated three times with similar results. **(c)** Spectroscopic analysis of NADH oxidation by human PYROXD1. Data points represent the mean ± SEM of three independent replicates. Solid lines represent a single-exponential fit. **(d)** In vitro pull-down experiment with recombinant PYROXD1 mutants and immobilized StrepII-GFP-RTCB in the presence of NADH. Strep-Tactin beads were washed to remove unbound PYROXD1, and bound proteins were analyzed by SDS-PAGE and Coomassie blue staining. This experiment was repeated three times with similar results.

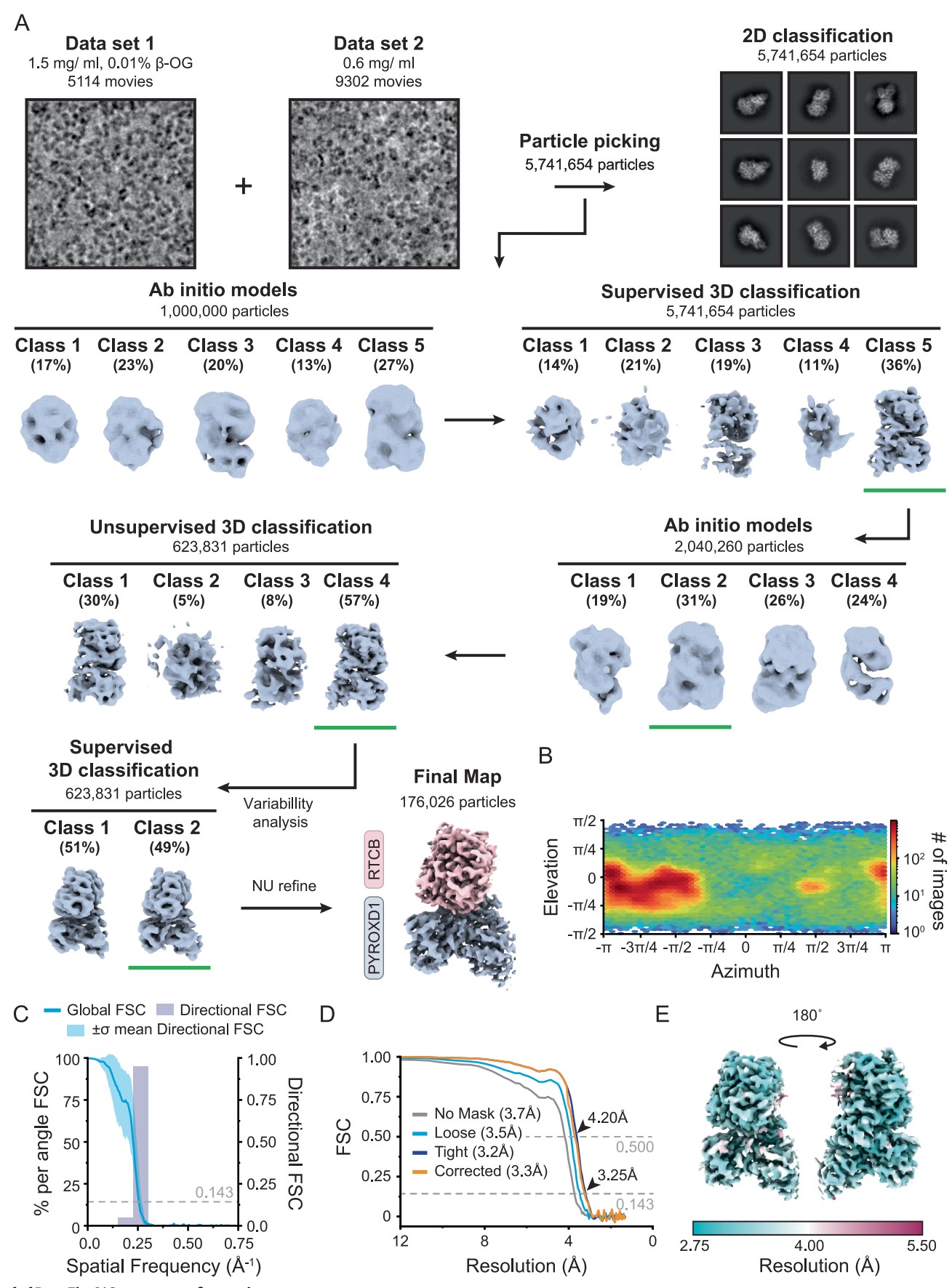

**Extended Data Fig. 2 | See next page for caption.**

**Extended Data Fig. 2 | Cryo-EM processing workflow for the human RTCB-PYROXD1 complex. (a)** Cryo-EM processing workflow for the human RTCB-PYROXD1 complex. **(b)** Euler diagram showing orientation distribution of final cryo-EM reconstruction. **(c)** Directional Fourier Shell Correlation (FSC) plot representing 3D resolution anisotropy in the reconstructed map. Blue line represents Mean directional FSC +/− Standard Deviation (STD). **(d)** FCS determined from two independently refined half maps. The gold standard cut-off (FCS = 0.143) is marked with an arrow. **(e)** Local resolution estimation on the final cryo-EM density map of the human RTCB-PYROXD1 complex.

**Extended Data Fig. 3 | PYROXD1 ligand interaction maps. (a)** Interaction map of the FADH⁻ ligand in the RTCB-PYROXD1 complex. **(b)** Interaction map of the NAD⁺ ligand in the RTCB-PYROXD1 complex. Dotted lines depict hydrogen-bonding interactions.

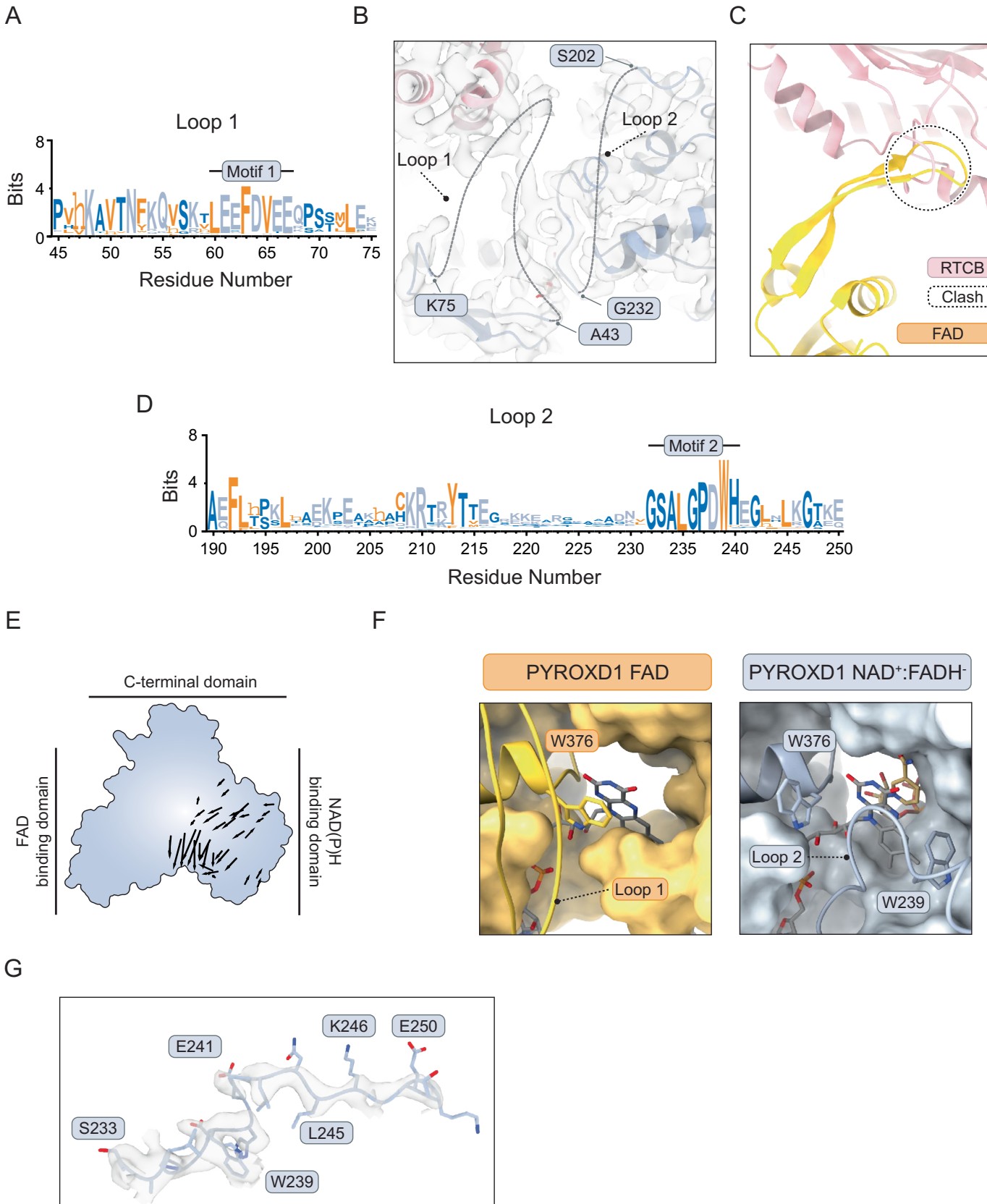

**Extended Data Fig. 4 | See next page for caption.**

**Extended Data Fig. 4 | Loops in PYROXD1 allosterically control its activity.**
**(a)** Sequence conservation analysis of loop 1 in PYROXD1. Figure was generated using WebLogo[38]. Black line indicates a conserved sequence motif (motif 1) in loop 1. **(b)** Detailed view of the projected paths of loop 1 and loop 2 in PYROXD1 (light blue) within the PYROXD1-RTCB complex, overlaid with cryo-EM map density (light grey). **(c)** Structural superposition of PYROXD1 bound to FAD (orange, PDB: 6ZK7)[19] and PYROXD1 in complex with RTCB (pink). Clashes are indicated by a dotted circle. **(d)** Sequence conservation analysis of loop 2 in PYROXD1. Black line indicates a conserved sequence motif (motif 2) in loop 2. **(e)** Vector map displaying the structural rearrangements in the NAD(P)H binding domain upon NAD(P)H binding and complex formation with RTCB. **(f)** Zoom-in views of the FAD binding site free PYROXD1 (orange, PDB: 6ZK7)[19] and PYROXD1 in complex with RTCB and NADH (light blue). **(g)** Conformation of the ordered part of Loop 2 in the PYROXD1-RTCB complex, overlaid with cryo-EM map density (light grey).

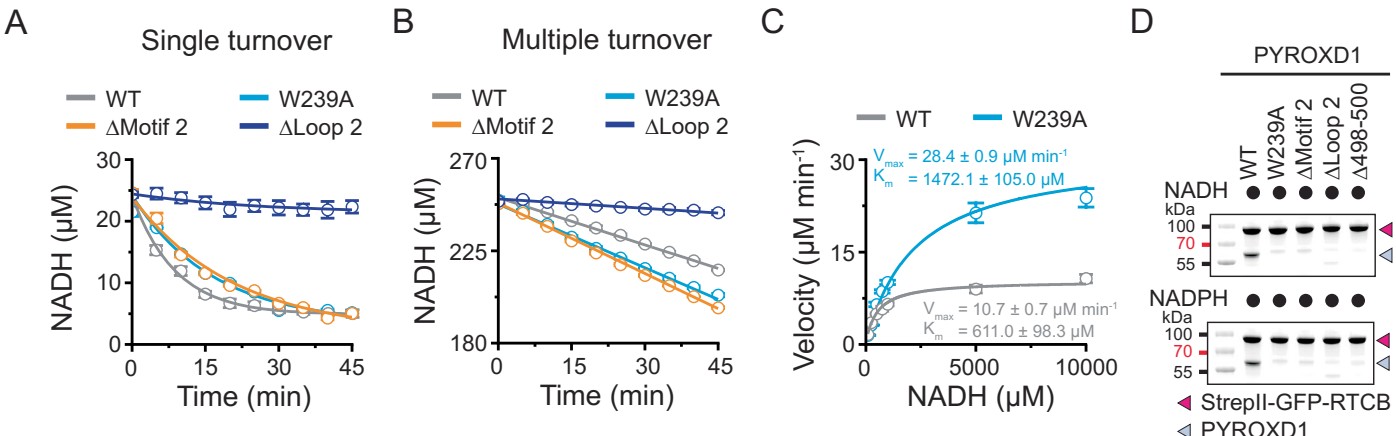

**Extended Data Fig. 5 | NADH turnover by PYROXD1. (a)** Single-turnover kinetics of NADH oxidation by human PYROXD1 variants under aerobic conditions. Data points represent the mean ± SEM of three independent replicates. Solid lines represent a single-exponential fit. **(b)** Multiple-turnover kinetics of NADH oxidation by human PYROXD1 variants under aerobic conditions. Data points represent the mean ± SEM of three independent replicates. Solid lines represent a linear fit. **(c)** Michealis-Menten analysis of NADH oxidation by human WT and W239A PYROXD1 under aerobic conditions. Data points represent the mean ± SEM of three independent replicates. Solid lines represent a Michaelis-Menten fit. **(d)** Co-precipitation of PYROXD1 mutants by immobilized StrepII-GFP-RTCB in the presence of NADH or NADPH. Bound proteins were analyzed by SDS-PAGE and Coomassie blue staining. This experiment was repeated three times with similar results.

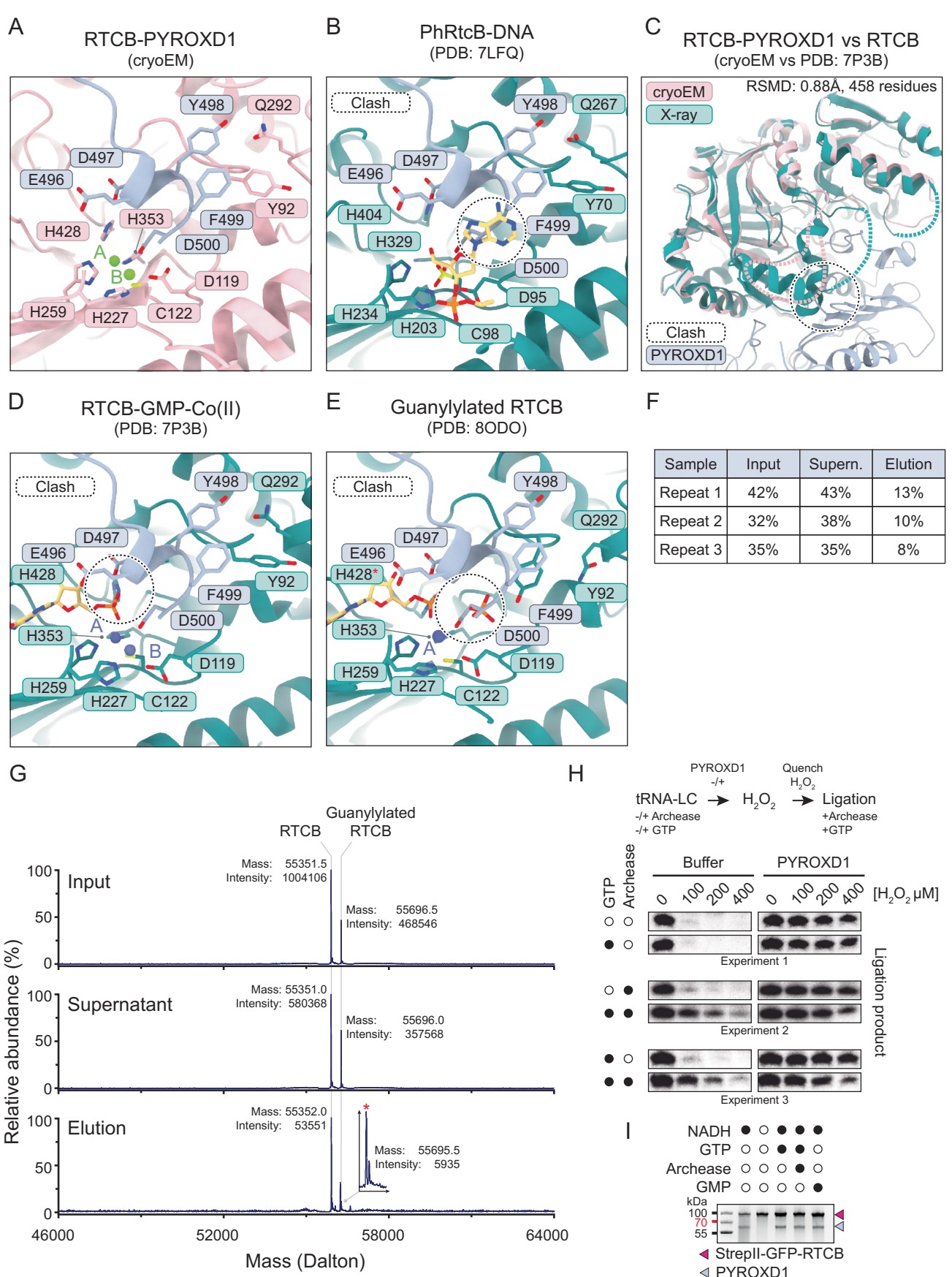

**Extended Data Fig. 6 | See next page for caption.**

**Extended Data Fig. 6 | Structural comparison of RTCB catalytic centers.**
**(a)** Detailed view of the interaction of the CTD of PYROXD1 with the catalytic center of RTCB. Bound magnesium ions are depicted as green spheres. **(b)** Structural superposition of the PYROXD1 CTD onto the catalytic cleft of DNA-bound PhRtcB (PDB: 7LFQ)[28]. Steric clashes are indicated by a dotted circle. **(c)** Structural superposition of the PYROXD1-RTCB complex and GMP-bound RTCB (PDB: 7P3B)[9]. **(d)** Structural superposition of the PYROXD1 CTD with the catalytic center of GMP-bound human RTCB (PDB: 7P3B)[9]. Clashes are indicated by a dotted circle. **(e)** Structural comparison of the position of the CTD of PYROXD1 in the catalytic center of guanylylated human RTCB (PDB: 8ODO)[10]. Clashes are indicated by a dotted circle. **(f)** Fraction of guanylylated RTCB in pull-down samples, as determined by LC-MS. Guanylylation levels are indicated for three independent replicates. **(g)** Representative mass spectra of the input, unbound (supernatant) and bound (elution) fractions. Mass and intensity of each peak are indicated. Red asterisk indicates a mass peak corresponding to GSH-modified RTCB. **(h)** In vitro ligation assay to determine the effects of guanylylation on the oxidative inactivation of RTCB. The ligation assay was performed in a three-step procedure, where RTCB was incubated with GTP and Archease (see labels on the left), followed by incubation with increasing concentrations of $H_2O_2$ in the presence or absence of PYROXD1. In the last steps all samples were supplemented with GTP and Archease to allow for multiple-turnover ligation of an RNA substrate. This experiment was repeated three times with similar results. **(i)** Co-precipitation of PYROXD1 by immobilized StrepII-GFP-RTCB in the presence of GTP, GMP or Archease. Strep-Tactin beads were washed to remove unbound PYROXD1, and bound proteins were analyzed by SDS-PAGE and Coomassie blue staining. This experiment was repeated three times with similar results.

|---|---|

# Reporting Summary

Please do not complete any field with "not applicable" or n/a.  Refer to the help text for what text to use if an item is not relevant to your study.
For final submission: please carefully check your responses for accuracy; you will not be able to make changes later.

## Statistics

For all statistical analyses, confirm that the following items are present in the figure legend, table legend, main text, or Methods section.

| n/a | Confirmed | |
|---|---|---|
| ☐ | ☒ | The exact sample size (*n*) for each experimental group/condition, given as a discrete number and unit of measurement |
| ☐ | ☒ | A statement on whether measurements were taken from distinct samples or whether the same sample was measured repeatedly |
| ☐ | ☒ | The statistical test(s) used AND whether they are one- or two-sided<br>*Only common tests should be described solely by name; describe more complex techniques in the Methods section.* |
| ☒ | ☐ | A description of all covariates tested |
| ☐ | ☒ | A description of any assumptions or corrections, such as tests of normality and adjustment for multiple comparisons |
| ☐ | ☒ | A full description of the statistical parameters including central tendency (e.g. means) or other basic estimates (e.g. regression coefficient) AND variation (e.g. standard deviation) or associated estimates of uncertainty (e.g. confidence intervals) |
| ☐ | ☒ | For null hypothesis testing, the test statistic (e.g. *F*, *t*, *r*) with confidence intervals, effect sizes, degrees of freedom and *P* value noted<br>*Give P values as exact values whenever suitable.* |
| ☒ | ☐ | For Bayesian analysis, information on the choice of priors and Markov chain Monte Carlo settings |
| ☒ | ☐ | For hierarchical and complex designs, identification of the appropriate level for tests and full reporting of outcomes |
| ☒ | ☐ | Estimates of effect sizes (e.g. Cohen's *d*, Pearson's *r*), indicating how they were calculated |

*Our web collection on statistics for biologists contains articles on many of the points above.*

## Software and code

Policy information about availability of computer code

| Data collection | EPU v2.9.0.1519REL, Gatan Digital micrograph v1.84.1282 |
|---|---|
| Data analysis | cryoSPARC v3.3.2, Coot v0.9.2, Phenix v1.20 including MolProbity, Origin Pro 2018, Prism GraphPad v9 |

For manuscripts utilizing custom algorithms or software that are central to the research but not yet described in published literature, software must be made available to editors and reviewers. We strongly encourage code deposition in a community repository (e.g. GitHub). See the Nature Portfolio guidelines for submitting code & software for further information.

## Data

Policy information about availability of data

All manuscripts must include a data availability statement. This statement should provide the following information, where applicable:
- Accession codes, unique identifiers, or web links for publicly available datasets
- A description of any restrictions on data availability
- For clinical datasets or third party data, please ensure that the statement adheres to our policy

The proteins used in this study are associated with the following UniProt accession codes: human Archease (Q8IWT0), human PYROXD1 (Q8WU10), and human RTCB (Q9Y3I0). Atomic coordinates and cryo-EM maps for the human RTCB-PYROXD1 complex (PDB: 8ORJ), EMDB: EMD-17127), have been deposited in the PDB and EMDB databases and are publicly available as of the date of publication. The models used for structural comparison are available as PDB IDs: 6ZK7, 7LFQ, 7P3B, 8ODO.

## Research involving human participants, their data, or biological material

Policy information about studies with human participants or human data. See also policy information about sex, gender (identity/presentation), and sexual orientation and race, ethnicity and racism.

| | |
|---|---|
| Reporting on sex and gender | N/A |
| Reporting on race, ethnicity, or other socially relevant groupings | N/A |
| Population characteristics | N/A |
| Recruitment | N/A |
| Ethics oversight | N/A |

Note that full information on the approval of the study protocol must also be provided in the manuscript.

# Field-specific reporting

Please select the one below that is the best fit for your research. If you are not sure, read the appropriate sections before making your selection.

☒ Life sciences          ☐ Behavioural & social sciences          ☐ Ecological, evolutionary & environmental sciences

For a reference copy of the document with all sections, see nature.com/documents/nr-reporting-summary-flat.pdf

# Life sciences study design

All studies must disclose on these points even when the disclosure is negative.

| | |
|---|---|
| Sample size | No statistics were employed to predetermine the sample size. The biochemical sample sizes were chosen (n>3) as such to ensure reproducibility of the observations. |
| Data exclusions | No data was excluded from the in vitro, cell-based and in vivo analyses. For cryo-EM experiments, consistent with standard protocols, picked particles that contributed to 2D classes and 3D reconstructions with lower resolution were removed |
| Replication | Information on replication of the experiments is reported in the figure legends. Experiments were executed at least three times to ensure reproducibility. All replication attempts showed similar results. |
| Randomization | This is not relevant for the experiments performed in this study because this study provides structural an biochemical chacaterisation of a protein complex and samples are not subject to systematic variation that demands randomisation, or were impossible to randomize because of the practical nature of these experiments. |
| Blinding | Blinding of investigators was not performed as there as no sample grouping. |

# Behavioural & social sciences study design

All studies must disclose on these points even when the disclosure is negative.

| | |
|---|---|
| Study description | N/A |
| Research sample | N/A |
| Sampling strategy | N/A |
| Data collection | N/A |
| Timing | N/A |
| Data exclusions | N/A |
| Non-participation | N/A |
| Randomization | N/A |

# Ecological, evolutionary & environmental sciences study design

All studies must disclose on these points even when the disclosure is negative.

| | |
|---|---|
| Study description | N/A |
| Research sample | N/A |
| Sampling strategy | N/A |
| Data collection | N/A |
| Timing and spatial scale | N/A |
| Data exclusions | N/A |
| Reproducibility | N/A |
| Randomization | N/A |
| Blinding | N/A |

Did the study involve field work?  ☐ Yes  ☒ No

## Field work, collection and transport

| | |
|---|---|
| Field conditions | N/A |
| Location | N/A |
| Access & import/export | N/A |
| Disturbance | N/A |

# Reporting for specific materials, systems and methods

We require information from authors about some types of materials, experimental systems and methods used in many studies. Here, indicate whether each material, system or method listed is relevant to your study. If you are not sure if a list item applies to your research, read the appropriate section before selecting a response.

## Materials & experimental systems

| n/a | Involved in the study |
|---|---|
| ☒ | ☐ Antibodies |
| ☐ | ☒ Eukaryotic cell lines |
| ☒ | ☐ Palaeontology and archaeology |
| ☒ | ☐ Animals and other organisms |
| ☒ | ☐ Clinical data |
| ☒ | ☐ Dual use research of concern |
| ☒ | ☐ Plants |

## Methods

| n/a | Involved in the study |
|---|---|
| ☒ | ☐ ChIP-seq |
| ☒ | ☐ Flow cytometry |
| ☒ | ☐ MRI-based neuroimaging |

## Antibodies

| | |
|---|---|
| Antibodies used | N/A |
| Validation | N/A |

April 2023

# Eukaryotic cell lines

Policy information about cell lines and Sex and Gender in Research

| | |
|---|---|
| Cell line source(s) | Sf9 insect cells, Thermo Scientific™, Cat#11496015 |
| Authentication | Sf9 Cell-line was guaranteed by the supplier Thermo Scientific™, Cat#11496015 and was therefore not authenticated |
| Mycoplasma contamination | Cells were not tested for mycoplasma contamination. |
| Commonly misidentified lines (See ICLAC register) | None of the cell-lines |

# Palaeontology and Archaeology

| | |
|---|---|
| Specimen provenance | N/A |
| Specimen deposition | N/A |
| Dating methods | N/A |

☐ Tick this box to confirm that the raw and calibrated dates are available in the paper or in Supplementary Information.

| | |
|---|---|
| Ethics oversight | N/A |

Note that full information on the approval of the study protocol must also be provided in the manuscript.

# Animals and other research organisms

Policy information about studies involving animals; ARRIVE guidelines recommended for reporting animal research, and Sex and Gender in Research

| | |
|---|---|
| Laboratory animals | N/A |
| Wild animals | N/A |
| Reporting on sex | N/A |
| Field-collected samples | N/A |
| Ethics oversight | N/A |

Note that full information on the approval of the study protocol must also be provided in the manuscript.

# Clinical data

Policy information about clinical studies

All manuscripts should comply with the ICMJE guidelines for publication of clinical research and a completed CONSORT checklist must be included with all submissions.

| | |
|---|---|
| Clinical trial registration | N/A |
| Study protocol | N/A |
| Data collection | N/A |
| Outcomes | N/A |

# Dual use research of concern

Policy information about dual use research of concern

## Hazards

Could the accidental, deliberate or reckless misuse of agents or technologies generated in the work, or the application of information presented in the manuscript, pose a threat to:

| No | Yes | |
|---|---|---|
| ☒ | ☐ | Public health |
| ☒ | ☐ | National security |
| ☒ | ☐ | Crops and/or livestock |
| ☒ | ☐ | Ecosystems |
| ☒ | ☐ | Any other significant area |

### Experiments of concern

Does the work involve any of these experiments of concern:

| No | Yes | |
|---|---|---|
| ☒ | ☐ | Demonstrate how to render a vaccine ineffective |
| ☒ | ☐ | Confer resistance to therapeutically useful antibiotics or antiviral agents |
| ☒ | ☐ | Enhance the virulence of a pathogen or render a nonpathogen virulent |
| ☒ | ☐ | Increase transmissibility of a pathogen |
| ☒ | ☐ | Alter the host range of a pathogen |
| ☒ | ☐ | Enable evasion of diagnostic/detection modalities |
| ☒ | ☐ | Enable the weaponization of a biological agent or toxin |
| ☒ | ☐ | Any other potentially harmful combination of experiments and agents |

# Plants

| | |
|---|---|
| Seed stocks | N/A |
| Novel plant genotypes | N/A |
| Authentication | |

# ChIP-seq

## Data deposition

☐ Confirm that both raw and final processed data have been deposited in a public database such as GEO.

☐ Confirm that you have deposited or provided access to graph files (e.g. BED files) for the called peaks.

| Data access links<br>*May remain private before publication.* | N/A |
|---|---|
| Files in database submission | N/A |
| Genome browser session<br>(e.g. UCSC) | N/A |

## Methodology

| | |
|---|---|
| Replicates | N/A |
| Sequencing depth | N/A |
| Antibodies | N/A |
| Peak calling parameters | N/A |
| Data quality | N/A |

| Software | N/A |
|---|---|

# Flow Cytometry

## Plots

Confirm that:

☐ The axis labels state the marker and fluorochrome used (e.g. CD4-FITC).

☐ The axis scales are clearly visible. Include numbers along axes only for bottom left plot of group (a 'group' is an analysis of identical markers).

☐ All plots are contour plots with outliers or pseudocolor plots.

☐ A numerical value for number of cells or percentage (with statistics) is provided.

## Methodology

| Sample preparation | N/A |
|---|---|
| Instrument | N/A |
| Software | N/A |
| Cell population abundance | N/A |
| Gating strategy | N/A |

☐ Tick this box to confirm that a figure exemplifying the gating strategy is provided in the Supplementary Information.

# Magnetic resonance imaging

## Experimental design

| Design type | N/A |
|---|---|
| Design specifications | N/A |
| Behavioral performance measures | N/A |

| Imaging type(s) | N/A |
|---|---|
| Field strength | N/A |
| Sequence & imaging parameters | N/A |
| Area of acquisition | N/A |

Diffusion MRI     ☐ Used     ☐ Not used

## Preprocessing

| Preprocessing software | N/A |
|---|---|
| Normalization | N/A |
| Normalization template | N/A |
| Noise and artifact removal | N/A |
| Volume censoring | N/A |

## Statistical modeling & inference

| Model type and settings | N/A |
|---|---|
| Effect(s) tested | N/A |

Specify type of analysis: ☐ Whole brain ☐ ROI-based ☐ Both

Statistic type for inference

N/A

(See Eklund et al. 2016)

Correction

N/A

## Models & analysis

| n/a | Involved in the study |
|---|---|
| ☒ ☐ | Functional and/or effective connectivity |
| ☒ ☐ | Graph analysis |
| ☒ ☐ | Multivariate modeling or predictive analysis |

Functional and/or effective connectivity

N/A

Graph analysis

N/A

Multivariate modeling and predictive analysis

N/A

