## [Peer Review File · Nature Structural & Molecular Biology]

Mechanistic basis for PYROXD1-mediated protection of the human tRNA ligase complex against oxidative inactivation

Corresponding Author: Professor Martin Jinek

Version 0:

Decision Letter:

9th Aug 2023

Dear Dr. Jinek,

Thank you again for submitting your manuscript "Mechanistic basis for PYROXD1-mediated protection of the human tRNA ligase complex against oxidative inactivation". I apologize for the delay in responding, which resulted from the difficulty in obtaining suitable referee reports. Nevertheless, we now have comments (below) from the 2 reviewers who evaluated your paper. In light of those reports, we remain interested in your study and would like to see your response to the comments of the referees, in the form of a revised manuscript.

You will see that Reviewer #1 raises concerns about the potential overinterpretation of the structural data in some regions, echoed by Reviewer #2 who raises concerns about the processing of the structural data, particularly aiming at improvement of the angular distribution and model statistics. Please be sure to address/respond to all concerns of the referees in full in a point-by-point response and highlight all changes in the revised manuscript text file. If you have comments that are intended for editors only, please include those in a separate cover letter.

We appreciate the revisions may require some time to perform, and we therefore expect to see your revised manuscript within 3 months. If you cannot send it within this time, please contact us to discuss an extension; we would still consider your revision, provided that no similar work has been accepted for publication at NSMB or published elsewhere.

Reporting Summary:

- that unprocessed scans are clearly labelled and match the gels and western blots presented in figures.
- that control panels for gels and western blots are appropriately described as loading or sample processing controls
- all images in the paper are checked for duplication of panels and for splicing of gel lanes.

Please note that all key data shown in the main figures as cropped gels or blots should be presented in uncropped form, with molecular weight markers. These data can be aggregated into a single supplementary figure item. While these data can be displayed in a relatively informal style, they must refer back to the relevant figures. These data should be submitted with the final revision, as source data, prior to acceptance, but you may want to start putting it together at this point.

SOURCE DATA: we request that authors provide, in tabular form, the data underlying the graphical representations used in figures. This is to further increase transparency in data reporting, as detailed in this editorial (<http://www.nature.com/nsmb/journal/v22/n10/full/nsmb.3110.html>). Spreadsheets can be submitted in excel format. Only one (1) file per figure is permitted; thus, for multi-paneled figures, the source data for each panel should be clearly labeled in the Excel file; alternately the data can be provided as multiple, clearly labeled sheets in an Excel file. When submitting files, the title field should indicate which figure the source data pertains to. We encourage our authors to provide source data at the revision stage, so that they are part of the peer-review process.

Data availability: this journal strongly supports public availability of data. All data used in accepted papers should be available via a public data repository, or alternatively, as Supplementary Information. If data can only be shared on request, please explain why in your Data Availability Statement, and also in the correspondence with your editor. Please note that for some data types, deposition in a public repository is mandatory - more information on our data deposition policies and available repositories can be found below:

<https://www.nature.com/nature-research/editorial-policies/reporting-standards#availability-of-data>

We require deposition of coordinates (and, in the case of crystal structures, structure factors) into the Protein Data Bank with the designation of immediate release upon publication (HPUB). Electron microscopy-derived density maps and coordinate data must be deposited in EMDDB and released upon publication. Deposition and immediate release of NMR chemical shift assignments are highly encouraged. Deposition of deep sequencing and microarray data is mandatory, and the datasets must be released prior to or upon publication. To avoid delays in publication, dataset accession numbers must be supplied with the final accepted manuscript and appropriate release dates must be indicated at the galley proof stage.

Link Redacted

Sincerely,
Sara

Sara Osman, Ph.D.
Associate Editor
Nature Structural & Molecular Biology

Referee expertise:

Referee #1: RNA structure and function

Referee #2: RNA-binding complexes, cryo-EM

Reviewers' Comments:

Reviewer #1:

Remarks to the Author:

The manuscript entitled "Mechanistic basis for PYROXD1-mediated protection of the human tRNA ligase complex against oxidative inactivation" by Loeff et al presents a structure-based mechanistic model for how PYROXD1 protects the catalytic activity of RTCB. The cryo-EM structure of RTCB in complex with PYROXD1 provides a unique insight into how NAD(P)H binding FAD control the intermolecular interaction between PYROXD1 and RTCB important to maintain tRNA ligase activity, making the combined biochemical and structural work interesting. The cryo-EM maps generally support the global orientation of the domains. However, many of the mechanistic conclusions that the authors draw also depend on the detailed interpretation of the structural data. The reviewer's major concern with the current manuscript is overinterpretation of the data in the final atomic model and inadequacies in the description of the structure model. Here are some specific points to help the authors improve the manuscript:

1. FADH and NAD binding pockets. The map for the two factors is somewhat reasonable, but the resolution is modest. The focus on hydrogen bonds at this resolution detracts from the reliability of the model. How were the "interaction maps" generated? What criteria were used to select the visualized residues? The authors seem to be missing some contacts and there are also close contacts between oxyanions of the phosphates and carbonyl oxygens. Can the authors describe the structure more thoroughly to clarify the driving factors that determine the observed conformation?
2. C-terminal helix of PYROXD1 binding the RTCB catalytic center. What is driving the recognition and affinity of the helix binding is not described in the text or the figure. In Fig 2B, the "interaction map" is incomplete and should also taken into consideration the modest resolution of the data. How were the side chains chosen for display in Fig 2B? What were the criteria for choosing certain dashed lines (presumably H-bonds)? In the text, the interactions between the terminal carboxy group of PYROXD1 and the backbone Phe118RTCB and Asp119RTCB are mentioned but not to the Lysine 375 of RTCB. Mg ions might also be lacking some of the ligands. The authors are urged to complete their structural analysis, and make interpretations taking into account the modest resolution.
3. Loops. Loop1 (PYROXD1 45-76) is not well-ordered and the map quality is poor in this region. Given that another disordered loop is nearby (202-237), how do the authors know that the provided model is valid for this loop? Can the authors perform any other experiments (eg. crosslinking)? They would be more helpful than loss of activity due to mutations (which the authors say they cannot make) anyway.
4. While the proposed model for W239 is attractive due to proximity, the map quality in this region is rather poor. Can the authors perform more experiments to exclude the possibility that W239 is not involved in another more direct interaction with RTCB? Does W239A mutation affect affinity for FAD independently of RTCB?
5. The description of an allosteric mechanism for Loop2 is unclear. Can they elaborate on how the distinct conformational changes are coupled? Without completely modeled loop(s) and without structural information for every state, the description should be clearer to illustrate what is observed and what is proposed.
6. Can the authors confirm how they determined the concentrations for single and multiple turnover conditions?
7. The authors use NADH and NADPH interchangeably. Could the authors include a discussion of how NADPH would be accommodated in their structure? Do the authors observe any biochemical differences for NADH vs NADPH for mutants near the binding site? Potentially, the extra phosphorylation could interfere with the structure.
8. The authors mention that Glu496 of PYROXD1 seems important to protect against a connective tissue disorder and myopathy. Glu496 of PYROXD1 seems to interact with Arg432 of RTCB. How do mutations of these residues affect the biochemical activity?
9. For the extended figures on cryo-EM data processing, please include angular distribution plots.
10. The model could be improved, especially to improve the clashes.

Minor points

Page 5 line 107. PYROXD1 residues 496-500 can't form "hydrophobic interactions" that are additional to Tyr498 or Phe499.

Reviewer #2:

Remarks to the Author:

In this manuscript, Loeff et al. provide structural and biochemical insights into the mechanism of how PYROXD1 protects the catalytic subunit of the tRNA ligase complex, RTCB, from oxidation.

Overall, this manuscript is very concisely written and the figures are very intuitive. The main result is the structure of the RTCB-PYROXD1 complex, which shows that PYROXD1 interacts directly with the active center of RTCB via its C-terminal tail to protect catalytic residues from oxidation. The model derived from the structural data is supplemented with biochemical data, which support the main conclusions of the paper.

I think this a very interesting story which provides fundamental new insights into how the tRNA ligase complex is protected from oxidative damage. However, I have some (mainly technical) concerns which I think need to be addressed before publication:

Major points:

- The authors state that the sample suffered from orientation bias. To overcome this, the authors have collected two datasets, one with and one without the addition of detergent, which they have merged for analysis. From the data provided by the authors, it is not clear whether and to which extent the orientation bias was overcome by this strategy. While the densities shown are certainly of good enough quality to model this complex, it is my impression that the maps still suffer from some residual orientation bias. Could the authors please provide plots of viewing angle distribution, and ideally also a 3D-FSC plot?
- In the cryo-EM workflow presented here, the two datasets with and without detergent are merged. Have the authors tried

processing the dataset with detergent independently to assess its effect on angular distribution? I am asking because in my experience, mixing data with and without (or with less) orientation bias has sometimes led to worse results than using only the dataset with without orientation bias, even if it that means fewer total particles. Another strategy I can strongly recommend is to skip 2D classification altogether during processing, and instead use only (supervised) 3D classification with one "good" and several "junk" classes. In my experience, this is a very powerful approach to recover rare views, which tend to be discarded in 2D classification (see for example PMID 32438371). If they have not done so, I would encourage the authors to explore these computational options to further improve the reconstruction. I believe this may greatly help strengthen some of the points made in the paper (see below points on densities).

- Lines 109-111: The level of detail at which interactions are described here is in my opinion not supported by the cryo-EM data. The density for D500 is discontinuous and somewhat ambiguous, which could be due to residual orientation bias.

- Lines 110-113: This statement seems to be in contrast with the observation that the D500A mutant binds basically the same as wildtype (Figure 2C). Similarly, in line 132 ff, the authors suggest that D500A and Δ 500 PYROXD1 may have reduced binding affinity. This could be easily tested for example by fluorescence anisotropy using the STREP-GFP-RTCB construct, which I think the authors should do if they wish to make this argument.

- Lines 137-147: The authors state here correctly that Loop1 appears to be (at best) only partially ordered in the RTCB-PYROXD1 complex. Nevertheless, the loop is modelled completely in the provided PDB file. In my opinion, the density for this region is highly ambiguous. In particular, it is not obvious to me how the authors traced the paths of Loop1 and Loop2 confidently, as both seem to differ to the previous crystal structure. Both regions are flagged in the PDB validation report for poor model to map fit in the EM density. I think this is a serious limitation of the data, which is not adequately represented in the current description and interpretation of the structure. Since these are key structural elements that are discussed in the paper, the authors should at the very least provide figures that show the fit of the model in this region to the density, discuss the limitations of the data, and consider a more conservative interpretation.

- Line 151: As above, Trp239 and the entire region have poor fit to density (see PDB validation report).

- The clash score for this model is relatively high, especially considering that the starting crystal structure models have substantially lower clash scores. Have the authors tried applying more stringent restraints to the models, for example by using the crystal structures as reference models?

Minor points:

- Figure 1C: The density is extremely difficult to see on-screen, and practically invisible on print-out

- Parts of the FAD ligand have discontinuous density. This is not a big issue because the ligand is positioned exactly as in a previous crystal structure with much better density. However, I think it should be stated in the methods that high-resolution models were fit and which models these were (PDB codes in text).

- Line 98: The list is missing E262, which is indicated in the referenced figures.

- Line 199: I think this sentence is missing "against".

- Extended Data Figure 4: The legend is missing a description for panel G.

Version 1:

Decision Letter:

28th Jun 2024

Dear Dr. Jinek,

Thank you again for submitting your manuscript "Mechanistic basis for PYROXD1-mediated protection of the human tRNA ligase complex against oxidative inactivation". I apologize for the delay in responding, which resulted from the difficulty in obtaining suitable referee reports. Nevertheless, we now have comments (below) from the 2 reviewers who evaluated your paper. In light of those reports, we remain interested in your study and would like to see your response to the comments of the referees, in the form of a revised manuscript.

You will see that the referees have a few remaining concerns. Please be sure to address/respond to all concerns of the referees in full in a point-by-point response and highlight all changes in the revised manuscript text file. If you have comments that are intended for editors only, please include those in a separate cover letter.

We expect to see your revised manuscript within 6 weeks. If you cannot send it within this time, please contact us to discuss an extension; we would still consider your revision, provided that no similar work has been accepted for publication at NSMB or published elsewhere.

Reporting Summary:

Please note that all key data shown in the main figures as cropped gels or blots should be presented in uncropped form, with molecular weight markers. These data can be aggregated into a single supplementary figure item. While these data can be displayed in a relatively informal style, they must refer back to the relevant figures. These data should be submitted with the final revision, as source data, prior to acceptance, but you may want to start putting it together at this point.

Data availability: this journal strongly supports public availability of data. All data used in accepted papers should be available via a public data repository, or alternatively, as Supplementary Information. If data can only be shared on request, please explain why in your Data Availability Statement, and also in the correspondence with your editor. Please note that for some data types, deposition in a public repository is mandatory - more information on our data deposition policies and available repositories can be found below:

<https://www.nature.com/nature-research/editorial-policies/reporting-standards#availability-of-data>

Link Redacted

Sincerely,
Sara

Sara Osman, Ph.D.
Senior Editor
Nature Structural & Molecular Biology

Reviewers' Comments:

Reviewer #1:

Remarks to the Author:

The revised manuscript by Loeff et al. is improved since the initial submission. However, there are still significant concerns that need to be addressed.

- While it is good to hear that the authors have improved the map quality, the resolution is still modest and the emphasis on hydrogen bonds and rotamers should be removed to avoid overinterpreting the structure data. For example, Figure 1 has changed significantly since the last time, and it may change with higher resolution data. The hydrogen bond array depends somewhat on interpretation. It may be best to remove the H-bonds.
- In the structure figures with partial views such as Fig 2B, a clearer definition of which residues are highlighted must be included in the Figure legend. Otherwise, it might seem subjective. Same in Fig 1D and 1E.
- The claim that the difference between 2-fold and 3-fold changes in V_{max} and K_m imply changes in K_d is rather circuitous with some room for uncertainty. Can the authors measure the change in affinity for NADH to accompany Extended Data Fig 5C?
- Co-precipitation experiments to measure the protein-protein affinity is at best semi-quantitative. Can they use a more quantitative binding assay, such as the one previously suggested by a reviewer?
- Lines 89-90. The authors say that the physiological substrate is likely to be NADPH. Can they provide an explanation for why they chose to use NADH instead of NADPH?
- For the response to Reviewer 1 point 8, the authors are recommended to include the data on disease mutations. Even if the mutations did not cause impressive biochemical changes *in vitro*, the data may be useful for future research on the disease mechanism.

Reviewer #2:

Remarks to the Author:

The authors have substantially revised this manuscript to address the reviewer comments. In particular, they have re-processed the cryo-EM data, performed additional experiments and revised the text. In my opinion, all major points have been adequately addressed and the manuscript can be published. I only have a few minor points left that I noticed:

- 1.) Line 160: I think Loop 1 needs to be introduced, i.e. stated that it corresponds to the loop described in the previous sentence.
- 2.) Line 167-169: This sentence ends abruptly and should be revised.
- 3.) Extended Data Figure 2: Please check the particle numbers, they are identical at several steps.

Version 2:

Decision Letter:

Our ref: NSMB-A47649B

13th Nov 2024

Dear Dr. Jinek,

Thank you for submitting your revised manuscript "Mechanistic basis for PYROXD1-mediated protection of the human tRNA ligase complex against oxidative inactivation" (NSMB-A47649B). Please accept our apologies for the delay in the decision. The manuscript has now been seen by the original referees and their comments are below. The reviewers find that the paper has improved in revision, and therefore we'll be happy in principle to publish it in Nature Structural & Molecular Biology, pending minor revisions to satisfy the referees' final requests and to comply with our editorial and formatting guidelines.

We are now performing detailed checks on your paper and will send you a checklist detailing our editorial and formatting requirements in about 2-3 weeks. Please do not upload the final materials and make any revisions until you receive this additional information from us.

To facilitate our work at this stage, it is important that we have a copy of the main text as a word file. If you could please send along a word version of this file as soon as possible, we would greatly appreciate it; please make sure to copy the NSMB account (cc'ed above).

Thank you again for your interest in Nature Structural & Molecular Biology. Please do not hesitate to contact me if you have any questions.

Sincerely,
Kat

Katarzyna Ciazynska, PhD
(she/her)
Senior Editor
Nature Structural & Molecular Biology
<https://orcid.org/0000-0002-9899-2428>

Reviewer #1 (Remarks to the Author):

This manuscript has addressed the concerns raised. Figure 1 has room for improvement as no context is shown for the interacting residues. However, I am fine with the changes they made to avoid overinterpreting the moderate resolution data.

Reviewer #2 (Remarks to the Author):

The authors have addressed all my points adequately, and I therefore support acceptance of this manuscript.

Version 3:

Decision Letter:

12th Feb 2025

Dear Dr. Jinek,

We are now happy to accept your revised paper "Mechanistic basis for PYROXD1-mediated protection of the human tRNA ligase complex against oxidative inactivation" for publication as a Article in Nature Structural & Molecular Biology.

Your paper will be published online soon after we receive proof corrections and will appear in print in the next available issue. You can find out your date of online publication by contacting the production team shortly after sending your proof corrections.

You may wish to make your media relations office aware of your accepted publication, in case they consider it appropriate to

organize some internal or external publicity. Once your paper has been scheduled you will receive an email confirming the publication details. This is normally 3-4 working days in advance of publication. If you need additional notice of the date and time of publication, please let the production team know when you receive the proof of your article to ensure there is sufficient time to coordinate. Further information on our embargo policies can be found here:
<https://www.nature.com/authors/policies/embargo.html>

Kind regards,
Florian

Dr Florian Ullrich
Senior Editor, Nature
Consulting Editor, Nature Structural & Molecular Biology
ORCID 0000-0002-1153-2040

Reviewer #1:

The manuscript entitled “Mechanistic basis for PYROXD1-mediated protection of the human tRNA ligase complex against oxidative inactivation” by Loeff et al presents a structure-based mechanistic model for how PYROXD1 protects the catalytic activity of RTCB. The cryo-EM structure of RTCB in complex with PYROXD1 provides a unique insight into how NAD(P)H binding FAD control the intermolecular interaction between PYROXD1 and RTCB important to maintain tRNA ligase activity, making the combined biochemical and structural work interesting. The cryo-EM maps generally support the global orientation of the domains. However, many of the mechanistic conclusions that the authors draw also depend on the detailed interpretation of the structural data. The reviewer’s major concern with the current manuscript is overinterpretation of the data in the final atomic model and inadequacies in the description of the structure model. Here are some specific points to help the authors improve the manuscript:

We thank the Reviewer for the positive feedback as well as the constructive comments and suggestions to improve the manuscript.

1. FADH and NAD binding pockets. The map for the two factors is somewhat reasonable, but the resolution is modest. The focus on hydrogen bonds at this resolution detracts from the reliability of the model. How were the “interaction maps” generated? What criteria were used to select the visualized residues? The authors seem to be missing some contacts and there are also close contacts between oxyanions of the phosphates and carbonyl oxygens.

We thank the Reviewer for carefully checking our structural data. In the meantime, we have reprocessed the cryo-EM data by bypassing 2D classification and using a supervised 3D classification strategy to obtain a better density map which allowed us to rebuild and refine the atomic model of the proteins and the bound ligands. Consequently, we have been able to review and revise the interaction network of PYROXD1 with the FADH⁻ and NAD⁺ ligands (**Fig. 1D, E**). While doing so, we applied a more stringent distance cutoff (3.35 Å) for defining potential hydrogen bonding interactions and also considered relative orientations of the interacting groups. In the respective figure panels, we depict contacts that we deem relevant to the specific recognition of the ligands. There are additional, mostly hydrophobic/van der Waals, contacts but we choose not to depict them to preserve clarity of the figures. For reference, we provide results of Ligplot analysis of the current atomic model below. Please note that Ligplot does not show hydrogen bonding interactions with the diphosphate moiety of NAD⁺, even though these groups are within hydrogen bonding distance limits. This is likely to be caused by suboptimal placement of the ligand in the atomic model given the limited resolution of the maps and the resulting suboptimal orientation of the hydrogen bond donor

and acceptor groups. Nevertheless, we show these interactions as they are highly conserved in other NAD(P)H oxidase enzymes.

Can the authors describe the structure more thoroughly to clarify the driving factors that determine the observed conformation?

We thank the Reviewer for the suggestion. In the revised manuscript, we now include a more thorough description (pages 6-7, line 153-198) and additional/revised figure panels (**Fig. 3B**, **Extended Data Fig. 4A-G**) to explain the conformational changes observed upon NADH binding and RTCB complex formation. In the previously determined crystal structure of FAD-bound PYROXD1 in the absence of NAD(P)H and RTCB (Asanovic et al., Mol Cell, 2021; PDB: 6ZK7), Loop 1 adopts a well-defined conformation. In our structure of the RTCB-PYROXD1 complex, Loop 1 becomes partially disordered and only traces of it are visible in the cryo-EM density map (**Extended Data Fig. 4B**). Nevertheless, it is clear that the loop is repositioned otherwise it would sterically clash with bound RTCB (**Extended Data Fig. 4C**). In turn, PYROXD1 Loop 2, which is disordered in the NADH-free crystal structure, undergoes partial ordering upon NADH and RTCB binding, reinforced by aromatic (side-on) stacking of Trp239 and the flavin ring of FADH-. This re-structuring appears to be driven by conformational changes in the NAD(P)H binding domain of PYROXD1 (**Extended Data Fig. 4E**) and in turn mediates the displacement of Loop 1 to enable interaction with RTCB. To illustrate the conformational changes in the NAD(P)H binding domain of PYROXD1, we now include a

vector representation of the conformational rearrangement (**Extended Data Fig. 4E**) and side-by-side comparisons of Loop1 and Loop 2 in the PYROXD1 and RTCB-PYROXD1 structures (**Fig. 3B and Extended Data Fig. 4F**).

2. C-terminal helix of PYROXD1 binding the RTCB catalytic center. What is driving the recognition and affinity of the helix binding is not described in the text or the figure. In Fig 2B, the “interaction map” is incomplete and should also taken into consideration the modest resolution of the data. How were the side chains chosen for display in Fig 2B? What were the criteria for choosing certain dashed lines (presumably H-bonds?)? In the text, the interactions between the terminal carboxy group of PYROXD1 and the backbone Phe118^{RTCB} and Asp119^{RTCB} are mentioned but not to the Lysine 375 of RTCB. Mg ions might also be lacking some of the ligands. The authors are urged to complete their structural analysis, and make interpretations taking into account the modest resolution.

We have used a supervised 3D classification approach to process the cryo-EM data and obtained an improved map, which allowed us to review the atomic interactions of the C-terminal helix of PYROXD1 with the active center of RTCB1. We have now revised the corresponding figure to depict all amino acid residues in the C-terminal helix of PYROXD1 and surrounding RTCB residues – we now show the interaction interface in two orientations as two panels (**Fig. 2B**). Dashed lines indicate either potential hydrogen bonding interactions (distance cutoff of 3.35 Å) or inner-sphere coordination of divalent metal cations. Please note that the coordination spheres of ions A (tetrahedral) and B (octahedral geometry) are incomplete as the remaining positions are presumably occupied by water molecules, which are not observed in the cryo-EM maps at the current resolution of the data but have been observed in the crystallographic structures of human RTCB. The C-terminal carboxylate indeed contacts the backbone amide group of Asp119^{RTCB}. However, Lys375^{RTCB} is too far for a hydrogen bonding/salt bridge contact with the C-terminal carboxylate, as the sidechain points away and makes contacts with the backbone carbonyl of Tyr92^{RTCB}.

To investigate the molecular determinants of RTCB-PYROXD1 binding, we have generated additional PYROXD1 mutant proteins and tested their binding to RTCB as well as their protective activities (**Fig. 2C and 2D**). Firstly, we examined a series of C-terminal truncations, removing one residue at a time. Here, we observe that removal of the C-terminal residue Asp500^{PYROXD1} alone is not sufficient to abolish binding to RTCB. Loss of RTCB binding is observed only once both Phe499^{PYROXD1} and Asp500^{PYROXD1} are deleted. Secondly, we have examined individual alanine substitutions of residues 497-500. This series of mutants reveals that individual mutations of the two aromatic residues in the C-terminal helix (Phe499^{PYROXD1} or Tyr498^{PYROXD1}) strongly reduce RTCB binding, while substitution of Asp497^{PYROXD1} does not affect RTCB interaction. Mutation of the C-terminal aspartate results in a slight but

reproducible reduction of RTCB binding, contrasting the result obtained when the C-terminal residue is deleted, where no effect was observed.

Based on these results, we conclude that the aromatic residues in the C-terminal helix of PYROXD1 are major contributors to the interaction with RTCB in terms of binding affinity, presumably via hydrophobic/stacking interactions with Phe62^{RTCB}, Tyr92^{RTCB} and Phe118^{RTCB}. However, as the PYROXD1-RTCB interaction is itself dependent on the presence of divalent cations, it is possible that the interaction of the C-terminal Asp500 sidechain with the divalent cation B contributes to the binding. In this context, it is also conceivable that the D500A and Δ 500 mutations lead to restructuring of the RTCB-PYROXD1 interface that preserves both RTCB and divalent cation binding, as both PYROXD1 mutants still contain a free carboxylate group at their C-termini, which could functionally replace the sidechain of Asp500 in contacting the divalent metal ion at position B. Another possible explanation for the observed divalent metal dependence of the RTCB-PYROXD1 interaction is that the cations are required to ensure electrostatic potential complementarity of the PYROXD1 and RTCB molecular interaction surfaces.

In general, loss or reduction of RTCB binding also correlates with loss of protective activity (**Fig. 2D**). However, we observe that the protection is reduced also for the Δ 500 mutant, indicating that binding is not in itself sufficient for RTCB protection. This observation further suggests that the C-terminal residue contributes to the protective activity of PYROXD1, possibly by directly shielding the active site due to its interaction with the divalent metals, thereby preventing ROS access to the oxidation-prone residue Cys122.

3. Loops. Loop1 (PYROXD1 45-76) is not well-ordered and the map quality is poor in this region. Given that another disordered loop is nearby (202-237), how do the authors know that the provided model is valid for this loop? Can the authors perform any other experiments (eg. crosslinking)? They would be more helpful than loss of activity due to mutations (which the authors say they cannot make) anyway.

We have made efforts to improve the quality of the EM maps by reprocessing of the data, using a supervised 3D classification approach. While this approach improved the overall map quality and allowed us to model parts of Loop 2 more accurately, the conformation Loop 1 remains poorly defined (**Extended Data Fig. 4B**), even though its general location at the RTCB-PYROXD1 interface is unambiguous. In light of the Reviewer's comments, we have decided to remove loop 1 from the structural model. In the revised manuscript we have now revised **Fig. 3B** to show order-disorder transitions in Loop 1 and Loop 2. We complement this figure with additional panels in **Extended Data Fig. 4**, showing the trace density of Loop1 in

the cryo-EM map (**Extended Data Fig. 4B**) and illustrating the steric clash with RTCB if the loop remained in the position observed in the free PYROXD1 structure (**Extended Data Fig. 4C**). Accordingly, we have revised the manuscript text to state that Loop 1 becomes disordered upon complex formation (page 6, lines 156-162):

“In the free PYROXD1 crystal structure¹⁹, a highly conserved loop within the FAD binding domain (residues Pro45^{PYROXD1}-Arg76^{PYROXD1}), which contains an invariant sequence motif (Motif 1), is well structured and positioned away from the FAD co-factor (**Fig. 3B left, Extended data Fig. 4A**). In the RTCB-PYROXD1 complex, the conformation of Loop 1 is less well defined due to structural disordering; however, residual density in the cryo-EM map suggests that the loop shifts towards the PYROXD1 active center to avoid a steric clash with RTCB (**Extended data Fig. 4B-C**).”

To validate our structural observations, we have made considerable efforts to map the PYROXD1-RTCB interaction using crosslinking-coupled mass spectrometry (XL-MS). We performed crosslinking reactions with PYROXD1 alone or in a mixture with RTCB, both in presence and absence of NADH. We observed multiple crosslinks between Loop 1 and Loop 2, both in the absence of RTCB as well as in its presence. Notably, the crosslinking patterns appear to change in the presence of RTCB and NADH, as compared to free PYROXD1, as new cross-links become detectable. We provide a schematic summary of the XL-MS data below for the Reviewer’s reference.

Overall, these data are consistent with our structural observations and suggest that conformational restructuring of the two loops occurs upon NAD(P)H binding and in the presence of RTCB. Furthermore, these results also suggest that free PYROXD1 has a certain degree of conformational flexibility in the Loop1 and Loop 2 regions. However, further extensive analysis would be required to obtain high-quality, quantitative data. We believe that this analysis would provide only an incremental advance in our understanding of the mechanism and is not justified in terms of the extra cost in time and resources. For that reason, we would prefer not including the XL-MS in the manuscript and ask the Reviewers for their understanding.

4. While the proposed model for W239 is attractive due to proximity, the map quality in this region is rather poor. Can the authors perform more experiments to exclude the possibility that W239 is not involved in another more direct interaction with RTCB? Does W239A mutation affect affinity for FAD independently of RTCB?

Reprocessing of our cryo-EM datasets yielded an improved map that shows that Trp239^{PYROXD1} is part of a short helical segment that docks in a hydrophobic pocket adjacent to the FADH⁻ ligand. The sidechain of Trp239 makes a side-on aromatic stacking interaction with the flavin ring system. In the revised manuscript, we have further expanded our kinetic analysis of this mutant by performing Michaelis-Menten analysis, which shows that both the K_m and v_{max} increase when upon W239A mutation (**Extended Data Fig. 5C**), indicating that the binding of NADH is weakened, while the catalytic turnover is increased. This latter is possibly due to an increase in the dissociation rate of the NAD(P)⁺ product, which is likely the rate limiting step (as judged by the differences in the observed rate constants from the single- and multiple-turnover kinetics). This suggests that Trp239 is involved in modulating both NADH binding as well as its catalytic turnover, consistent with the previous biochemical data presented in the initially submitted version of the manuscript. The results of the extended kinetic analysis are also shown below for Reviewer's reference.

PYROXD1 natively co-purifies with endogenous FAD present in the cell, precluding us from directly measuring the affinity of PYROXD1 for FAD. To estimate the amount of FAD bound to PYROXD1, we compared the absorption spectra of WT PYROXD1 and the W239A mutant at nominally equal protein concentrations. As shown below for the Reviewer's reference, the proteins show near-identical absorption spectra with two peaks at 370 and 450 nm, which are characteristic for flavoproteins and flavin cofactors, suggesting that both the WT and W239A are bound with near-identical amounts of the FAD cofactor.

5. The description of an allosteric mechanism for Loop2 is unclear. Can they elaborate on how the distinct conformational changes are coupled? Without completely modeled loop(s) and without structural information for every state, the description should be clearer to illustrate what is observed and what is proposed.

Loop 2 of PYROXD1, which is disordered in the FAD-bound crystal structure (PDB: 6ZK7), undergoes partial ordering upon NADH and RTCB binding, reinforced by aromatic (side-on) stacking of Trp239 and the flavin ring of FADH⁻. This re-structuring is enabled by conformational changes in the NAD(P)H binding domain of PYROXD1. Loop 2 restructuring in turn leads to the displacement of Loop 1, which exposes the RTCB binding site. To illustrate the conformational changes in the NAD(P)H binding domain of PYROXD1, we have now included a vector representation of the conformational rearrangement (**Extended Data Fig. 4E**) and side-by-side comparisons of Loop1 and Loop 2 in the PYROXD1 and RTCB-PYROXD1 structures (**Fig. 3B** and **Extended Data Fig. 4F**). We have expanded the description of the conformational changes in the text, stating the following (page 6, lines 165-180) “The structure of the RTCB-PYROXD1 complex further reveals that the NAD(P)H binding domain of PYROXD1 undergoes a structural rearrangement upon NADH binding and RTCB recruitment (**Extended Data Fig. 4E**). A long loop (residues Ala197^{PYROXD1}-Glu251^{PYROXD1}, hereafter referred to as Loop 2), which connects the NAD(P)H binding and C-terminal domains and is structurally disordered in the NAD(P)H-free crystal structure of PYROXD1¹⁹. In the RTCB-PYROXD1 complex, segments of Loop 2 become ordered via interactions with the restructured NAD(P)H binding domain (**Fig. 3B right** and **Extended data Fig. 4E-G**), thereby

positioning a conserved motif (Motif 2) within Loop 2 (**Extended data Fig. 4D**) such that the side chain of Trp239^{PYROXD1} contacts the flavin ring of FADH⁻ via side-on aromatic stacking (**Fig. 3B** and **Extended data Fig. 4F**). In turn, Trp376^{PYROXD1}, which caps the flavin ring of FAD in the crystal structure of free PYROXD1 and restricts its access to electron acceptors¹⁹, is displaced away in the RTCB-PYROXD1 complex (**Extended data Fig. 4F**). As this would otherwise lead to a steric clash with Loop 1 in the absence of its restructuring, these observations suggest that NAD(P)H-dependent allosteric rearrangement of Loop 2 induces, via Trp376^{PYROXD1}, the disordering of Loop 1, thereby exposing of the RTCB interaction interface. Furthermore, the observed displacement of Trp376^{PYROXD1} upon allosteric restructuring of Loop 2 also suggests that Loop 2 modulates the NAD(P)H oxidase activity of PYROXD1 by regulating access to electron acceptors.”

6. Can the authors confirm how they determined the concentrations for single and multiple turnover conditions?

The criteria for both single- and multiple-turnover conditions were based on the previously established parameters in our initial characterization of PYROXD1 (Asanovic et al., Mol Cell, 2021). Under the chosen experimental conditions (25 μ M NADH starting concentration for single-turnover and 250 μ M NADH for multiple-turnover, with a PYROXD1 concentration of 25 μ M) we observe a single-exponential decay pattern for single-turnover and a linear decay for multiple-turnover, indicating that the NADH concentrations used in the assay fall within the appropriate ranges.

7. The authors use NADH and NADPH interchangeably. Could the authors include a discussion of how NADPH would be accommodated in their structure? Do the authors observe any biochemical differences for NADH vs NADPH for mutants near the binding site? Potentially, the extra phosphorylation could interfere with the structure.

In our previous study on PYROXD1 (Asanovic et al., Mol Cell, 202), we extensively examined the effects of NADH and NADPH through pull-down assays, spectroscopy, and molecular dynamics simulations. However, no significant differences were observed between

NADH and NADPH in these analyses. In the revised manuscript, we have extended our analysis with additional pull-down experiments that probe mutations in Loop 2 and the C-terminal domain (**Extended data Fig. 5D**, also shown below for Reviewer's reference). Consistent with our previous study, we did not observe any discernible difference between the effects of NADH and NADPH on the RTCB-PYROXD1 interaction.

In addition to our biochemical analysis, we have reviewed the structural model of the PYROXD1-RTCB complex. The adenosyl moiety of NADH is inserted in a pocket between Thr317^{PYROXD1} and Lys176^{PYROXD1}, with no apparent base-specific interactions. The 2'-hydroxyl group of the adenosyl moiety of NADH is oriented towards the solvent and is not engaged in any protein contacts, suggesting it could accommodate a phosphate group of NADPH without any steric hindrance from the protein, in agreement with our experimental observations. We have included the following statement in the Results section of the revised manuscript:

p. 4, lines 86-87. "PYROXD1 is able to oxidize NADH and NADPH with comparable kinetics, although the physiological substrate is likely to be NADPH, given the nucleocytoplasmic localization of PYROXD1."

p. 4, lines 107-110: "The 2'-hydroxyl group of the adenosyl moiety of NADH is oriented towards the solvent and is not engaged in any protein contacts, suggesting that it could accommodate a phosphate group of NADPH without any steric hindrance from the protein, in agreement with previously measured enzyme kinetics data."

8. *The authors mention that Glu496 of PYROXD1 seems important to protect against a connective tissue disorder and myopathy. Glu496 of PYROXD1 seems to interact with Arg432 of RTCB. How do mutations of these residues affect the biochemical activity?*

The ΔGlu496 mutation has been shown to be associated with a connective tissue disorder and myopathy. To probe the effect of the mutation on the activity of PYROXD1 and its interaction with RTCB, we carried out RTCB co-precipitation and protection experiments with ΔGlu496 and E496A PYROXD1 mutant proteins, and show the results below for the Reviewer's reference. Neither the deletion nor the alanine substitution of Glu496 affected interaction with RTCB or its protection from oxidative inactivation under the chosen experimental conditions. Taken at face value, these results suggest that the disease-linked

mutation does not affect the activity of PYROXD1, at least in vitro. It is possible that the mutations lead to restructuring of the C-terminal domain of PYROXD1 that (to some extent) preserves RTCB binding and protection in vitro. Nevertheless, it is not inconceivable that the mutation has a much more pronounced effect by compromising PYROXD1 expression and/or stability in vivo, leading to a disease phenotype. Finally, it is possible that PYROXD1 has other functions and/or interacting partners in addition to RTCB and the disease phenotype is caused by the loss of that function or interaction. As the molecular basis for the pathogenic mutation is presently unclear, we have taken the decision to remove the reference to the Δ Glu496 mutation from the manuscript.

9. For the extended figures on cryo-EM data processing, please include angular distribution plots.

In the revised version of the manuscript, we have now included an angular distribution plot (Extended Data Fig. 2B) as well as a 3D-FSC plot (Extended Data Fig. 2C), and show the plots below for the Reviewer's reference.

10. The model could be improved, especially to improve the clashes.

We thank the Reviewer for this suggestion, we have rebuilt and re-refined our model against the re-processed map using the crystallographic structures as reference models for restrained refinement. This lowered the clash score by ~80%, from 28 to 5.

Minor points

Page 5 line 107. PYROXD1 residues 496-500 can't form "hydrophobic interactions" that are additional to Tyr498 or Phe499.

In the revised manuscript, we have corrected the text (p. 5-6, lines 116-120) to state: "The interaction is centered on the C-terminal tail of PYROXD1 (residues Ile493–Asp500), which is unique to PYROXD1 and absent from other related NAD(P)H oxidases. The C-terminal tail is structurally disordered in free PYROXD1 but adopts an alpha-helical conformation upon interaction with RTCB, as it inserts into the catalytic center of RTCB and forms hydrophobic interactions with RTCB via Ile493^{PYROXD1}, Ile495^{PYROXD1}, Tyr498^{PYROXD1} and Phe499^{PYROXD1} (Fig. 2B)."

Reviewer #2:

In this manuscript, Loeff et al. provide structural and biochemical insights into the mechanism of how PYROXD1 protects the catalytic subunit of the tRNA ligase complex, RTCB, from oxidation. Overall, this manuscript is very concisely written and the figures are very intuitive. The main result is the structure of the RTCB-PYROXD1 complex, which shows that PYROXD1 interacts directly with the active center of RTCB via its C-terminal tail to protect catalytic residues from oxidation. The model derived from the structural data is supplemented with biochemical data, which support the main conclusions of the paper. I think this a very interesting story which provides fundamental new insights into how the tRNA ligase complex is protected from oxidative damage. However, I have some (mainly technical) concerns which I think need to be addressed before publication:

We thank the Reviewer for the positive assessment of our work and the constructive comments which have helped us improve the manuscript.

Major points:

- The authors state that the sample suffered from orientation bias. To overcome this, the authors have collected two datasets, one with and one without the addition of detergent, which they have merged for analysis. From the data provided by the authors, it is not clear whether and to which extent the orientation bias was overcome by this strategy. While the densities shown are certainly of good enough quality to model this complex, it is my impression that the maps still suffer from some residual orientation bias. Could the authors please provide plots of viewing angle distribution, and ideally also a 3D-FSC plot?

In the revised version of the manuscript, we have included an angular distribution plot (Extended Data Fig. 2B) as well as a 3D-FSC plot (Extended Data Fig. 2C), and shown below for the Reviewer's reference.

- In the cryo-EM workflow presented here, the two datasets with and without detergent are merged. Have the authors tried processing the dataset with detergent independently to assess its effect on angular distribution? I am asking because in my experience, mixing data with and without (or with less) orientation bias has sometimes led to worse results than using only the dataset with without orientation bias, even if it that means fewer total particles. Another

strategy I can strongly recommend is to skip 2D classification altogether during processing, and instead use only (supervised) 3D classification with one “good” and several “junk” classes. In my experience, this is a very powerful approach to recover rare views, which tend to be discarded in 2D classification (see for example PMID 32438371). If they have not done so, I would encourage the authors to explore these computational options to further improve the reconstruction. I believe this may greatly help strengthen some of the points made in the paper (see below points on densities).

We thank the Reviewer for the suggestions concerning the cryo-EM data processing strategy. We attempted processing each dataset separately but this introduced bias with distinct orientations, as illustrated in the 2D classes below, and resulted in distorted maps. The only viable option for obtaining an interpretable map was to merge the two datasets.

In our efforts to improve the map quality, we have now reprocessed the merged data, adopting a supervised 3D classification strategy suggested by the Reviewer and obtained a qualitatively better map that enabled more accurate modelling of the complex. This is reflected in improved statistics of the model.

- Lines 109-111: The level of detail at which interactions are described here is in my opinion not supported by the cryo-EM data. The density for D500 is discontinuous and somewhat ambiguous, which could be due to residual orientation bias.

The newly generated maps have helped to resolve this issue, allowing us to visualize the interaction interface of PYROXD1 and RTCB more clearly. We have updated **Fig. 2B** to show all contacts that we consider to be within hydrogen bonding distances (plus divalent cation coordination interactions).

- Lines 110-113: This statement seems to be in contrast with the observation that the D500A mutant binds basically the same as wildtype (Figure 2C). Similarly, in line 132, the authors suggest that D500A and Δ 500 PYROXD1 may have reduced binding affinity. This could be easily tested for example by fluorescence anisotropy using the STREP-GFP-RTCB construct, which I think the authors should do if they wish to make this argument.

We thank the Reviewer for bringing up this point. We acknowledge that the D500A and Δ 500 PYROXD1 mutant proteins appear to have the same ability to bind RTCB under the tested experimental conditions. Based on extended mutational analysis of the residues in the C-terminal tail of PYROXD1 using pull-down experiments (see response to Reviewer #1), we now conclude the aromatic sidechains of Tyr498 and Phe499 are main structural determinants underpinning the affinity between PYROXD1 and RTCB. Nevertheless, the RTCB-PYROXD1 interaction is dependent on the presence of divalent cations, suggesting that the interactions of the C-terminal tail with the two cations in the RTCB active center might contribute to the interaction. The mutations D500A and Δ 500 PYROXD1 do not appear to perturb the binding (**Fig. 2C**), although we do observe a minor reduction in the protective activity of PYROXD1 for the Δ 500 PYROXD1 mutant (**Fig. 2D**). It is not inconceivable that the mutations lead to restructuring of the RTCB-PYROXD1 interface that preserves RTCB and divalent cation binding, as both the D500A and Δ 500 PYROXD1 mutants still contain a free carboxylate group at their C-termini, which could potentially replace the sidechain of Asp500 in contacting the divalent metal ion at position B.

- Lines 137-147: The authors state here correctly that Loop1 appears to be (at best) only partially ordered in the RTCB-PYROXD1 complex. Nevertheless, the loop is modelled completely in the provided PDB file. In my opinion, the density for this region is highly ambiguous. In particular, it is not obvious to me how the authors traced the paths of Loop1 and Loop2 confidently, as both seem to differ to the previous crystal structure. Both regions are flagged in the PDB validation report for poor model to map fit in the EM density. I think this is a serious limitation of the data, which is not adequately represented in the current description and interpretation of the structure. Since these are key structural elements that are discussed in the paper, the authors should at the very least provide figures that show the fit of the model in this region to the density, discuss the limitations of the data, and consider a more conservative interpretation.

We appreciate the Reviewer's point and agree that a more conservative interpretation would be better. This is also in light of the improved maps, which still show that the density corresponding to Loop 1 remains poorly ordered. Based on this, we have decided to remove Loop 1 from the structural model. We have revised the corresponding section of the manuscript (p. 6-7, lines 154-199) to describe the observed structural changes in the two loops, stating that Loop 1 undergoes becomes disordered in the RTCB-bound complex and that the conformational restructuring is necessary for RTCB binding as it would otherwise lead to steric clash. We also note that the improved map allowed us to model part of Loop 2 with a greater degree of confidence (**Extended data figure 4G**), allowing us to propose a mechanism for the allosteric control of RTCB binding by conformational rearrangement in the NAD(P)H binding domain of PYROXD1 in the presence of NAD(P)H, followed by Loop 2

restructuring. This enables Loop 1 to reposition, thereby exposing the interface necessary for RTCB binding.

In the revised version of the manuscript, we now state (p. 6, lines 165-180):

“The structure of the RTCB-PYROXD1 complex further reveals that the NAD(P)H binding domain of PYROXD1 undergoes a structural rearrangement upon NADH binding and RTCB recruitment (**Extended Data Fig. 4E**). A long loop (residues Ala197^{PYROXD1}-Glu251^{PYROXD1}, hereafter referred to as Loop 2), which connects the NAD(P)H binding and C-terminal domains and is structurally disordered in the NAD(P)H-free crystal structure of PYROXD1¹⁹. In the RTCB-PYROXD1 complex, segments of Loop 2 become ordered via interactions with the restructured NAD(P)H binding domain (**Fig. 3B right and Extended Data Fig. 4E-G**), thereby positioning a conserved motif (Motif 2) within Loop 2 (**Extended Data Fig. 4D**) such that the side chain of Trp239^{PYROXD1} contacts the flavin ring of FADH- via side-on aromatic stacking (**Fig. 3B and Extended Data Fig. 4F**). In turn, Trp376^{PYROXD1}, which caps the flavin ring of FAD in the crystal structure of free PYROXD1 and restricts its access to electron acceptors¹⁹, is displaced away in the RTCB-PYROXD1 complex (**Extended Data Fig. 4F**). As this would otherwise lead to a steric clash with Loop 1 in the absence of its restructuring, these observations suggest that NAD(P)H-dependent allosteric rearrangement of Loop 2 induces, via Trp376^{PYROXD1}, the disordering of Loop 1, thereby exposing of the RTCB interaction interface. Furthermore, the observed displacement of Trp376^{PYROXD1} upon allosteric restructuring of Loop 2 also suggests that Loop 2 modulates the NAD(P)H oxidase activity of PYROXD1 by regulating access to electron acceptors.”

- *Line 151: As above, Trp239 and the entire region have poor fit to density (see PDB validation report).*

Reprocessing of our cryo-EM datasets yielded an improved map that shows that Trp239 is part of a short helical segment that docks in a hydrophobic pocket adjacent to the FAD ligand (**Fig. 3B, Extended Data Fig. 4F**). The side chain of Trp239 makes a side-on aromatic stacking interaction with the flavin ring system. The structural data is consistent with the biochemical/kinetics data obtained for the W293A mutant. Please also see our response to comment 4 of reviewer 1.

- *The clash score for this model is relatively high, especially considering that the starting crystal structure models have substantially lower clash scores. Have the authors tried applying more stringent restraints to the models, for example by using the crystal structures as reference models?*

We thank the Reviewer for this suggestion, we have re-refined our model against the re-processed map using the crystallographic structures as reference models for restrained refinement. This lowered the clash score by ~80%, from 28 to 5.

Minor points:

- Figure 1C: The density is extremely difficult to see on-screen, and practically invisible on print-out

We thank the Reviewer for this point. We have updated **Fig. 1C** to show the density at a lower level of transparency, such that the density is better visible on screen and on print-out.

- Parts of the FAD ligand have discontinuous density. This is not a big issue because the ligand is positioned exactly as in a previous crystal structure with much better density. However, I think it should be stated in the methods that high-resolution models were fit and which models these were (PDB codes in text).

In the improved map the FADH- ligand has continuous density that is in agreement with the crystal structure. For clarification, we have also included the following statement in the Methods section (Page 16, line 510-511):” The structural model of the RTCB-PYROXD1 was built in Coot (V0.9.2)³⁶, using crystallographic structures of RTCB⁹ and PYROXD1¹⁹ as starting models.”.

- Line 98: The list is missing E262, which is indicated in the referenced figures.

In the original version of the manuscript, E262 was erroneously shown to be within hydrogen-bonding distance of the adenine moiety. Based on the new maps obtained by reprocessing of the cryo-EM datasets, we have reviewed and revised the interaction network of PYROXD1 residues contacting the FADH⁻ and NAD⁺ ligands, concluding that E262 is not in direct contact with the NAD⁺.

- Line 199: I think this sentence is missing “against”.

We thank the Reviewer you for pointing out this error, we have corrected the sentence accordingly.

- Extended Data Figure 4: The legend is missing a description for panel G.

We have included the missing legend in the revised version of the manuscript.

Response to Reviewers' Comments:

Reviewer #1:

Remarks to the Author:

The revised manuscript by Loeff et al. is improved since the initial submission. However, there are still significant concerns that need to be addressed.

We thank the Reviewer for appreciating our revised manuscript and the additional constructive feedback.

- While it is good to hear that the authors have improved the map quality, the resolution is still modest and the emphasis on hydrogen bonds and rotamers should be removed to avoid overinterpreting the structure data. For example, Figure 1 has changed significantly since the last time, and it may change with higher resolution data. The hydrogen bond array depends somewhat on interpretation. It may be best to remove the H-bonds.

We appreciate the Reviewer's point. To avoid overinterpreting our structural results, we have revised the figures and removed the dashed lines that were previously used to indicate H-bonds.

- In the structure figures with partial views such as Fig 2B, a clearer definition of which residues are highlighted must be included in the Figure legend. Otherwise, it might seem subjective. Same in Fig 1D and 1E.

In the aforementioned figures, we have used a distance cut-off of 3.35 to show residues in the vicinity of the bound ligands in PYROXD1 (Fig. 1D-E) and in the active site of RTCB (Fig. 2B). We have revised the figure legends in the manuscript to mention this explicitly, as follows:

Figure 1D: "Interaction map of residues within a 3.35 Å distance from the FADH² ligand in the RTCB-PYROXD1 complex."

Figure 1E: "Interaction map of residues within a 3.35 Å distance from the NAD⁺ ligand in the RTCB-PYROXD1 complex."

Figure 2B: "Detailed view of the interactions of the CTD of PYROXD1 within a 3.35 Å distance from the catalytic center of RTCB."

- The claim that the difference between 2-fold and 3-fold changes in V_{max} and K_m imply changes in K_d is rather circuitous with some room for uncertainty. Can the authors measure the change in affinity for NADH to accompany Extended Data Fig 5C?

We thank the Reviewer for this suggestion. However, given that PYROXD1 constantly turns over NAD(P)H under aerobic conditions and it is not possible to measure the binding affinity for NAD(P)H directly under the experimental conditions used in this study.

We previously made several attempts to measure the substrate affinity using ITC, but unfortunately, these experiments did not yield meaningful results due to the heat changes caused by the constant enzymatic turnover. It would theoretically be possible to measure the binding under anaerobic conditions, but this would require the use of an anaerobic chamber setup, which we currently do not have access to. However, even then, we would not be able to measure the *true* affinity for NAD(P)H with methods such as ICT or SPR, because the enzyme rapidly forms the NADP⁺/FADH⁻ charge transfer complex under these conditions, as shown previously (Asanovic et al., Mol Cell, 2021).

K_m is a close approximation of the substrate dissociation constant K_D under conditions where dissociation of the enzyme-substrate complex is rapid relative to the catalytic turnover of the complex (i.e. $k_{-1} \gg k_2$). If this condition is not met, then K_m is always greater than true K_D - it is mathematically impossible for it to be smaller. In either case, K_m is thus a useful parameter to estimate substrate affinity insofar that it provides an upper limit on the true K_D for substrate binding, even if the latter cannot be measured experimentally.

- Co-precipitation experiments to measure the protein-protein affinity is at best semi-quantitative. Can they use a more quantitative binding assay, such as the one previously suggested by a reviewer?

We thank the Reviewer for the suggestion. As stated above, quantitative analysis of the RTCB-PYROXD1 interaction affinity is challenging due to its dependency on NAD(P)H and constant enzymatic turnover during the binding experiments. This rules out the use of ITC or SPR for biophysical measurements, which we are unable to perform under anaerobic conditions. As we do not aim to make strong quantitative claims about the effect of the mutations on the affinity of RTCB for PYROXD1, we believe that the Reviewer's request is not reasonable in view of the considerable effort required to realize these experiments and the expected added value of the results in the context of the overall conclusions of our study. Our main message is that the C-terminal tail of PYROXD1 is indispensable for RTCB binding, which is sufficiently evident from the semi-quantitative pull-down experiments.

- Lines 89-90. The authors say that the physiological substrate is likely to be NADPH. Can they provide an explanation for why they chose to use NADH instead of NADPH?

In our previous study on PYROXD1 (Asanovic et al, Mol Cell, 2021) , we extensively examined the effects of NADH and NADPH on PYROXD1 through a combination of *in vitro* pull-down assays, spectroscopy, and molecular dynamics simulations. However, no significant differences were observed between the effects of NADH and NADPH in these experiments. Given that NAD(P)H is less stable than NADH in solution, we chose to work with NADH for the reconstitution of the RTCB-PYROXD1 complex for cryo-EM studies, which were done under aerobic conditions. Additionally, cryo-EM sample preparation and optimization required large quantities of NADH (because NADH was included in size exclusion chromatography buffers), which is about tenfold cheaper than NAD(P)H. Considering these factors, we decided to perform all experiments with NADH for consistency.

- For the response to Reviewer 1 point 8, the authors are recommended to include the data on disease mutations. Even if the mutations did not cause impressive biochemical changes in vitro, the data may be useful for future research on the disease mechanism.

In the revised manuscript we have added the data on the $\Delta E496$ disease mutation in Extended data figure 1. We have included the following statement on Page 5, Lines 140-143: “Notably, deletion of Glu496 ($\Delta E496$) has recently been shown to be associated with a connective tissue disorder and myopathy²¹. However, the $\Delta E496$ PYROXD1 mutant protein retained its ability to interact with RTCB at near-wild-type levels in affinity co-precipitation experiments (**Extended data Fig. 1D**), suggesting that the pathology is not caused by loss of RTCB protection.”

Reviewer #2:

Remarks to the Author:

The authors have substantially revised this manuscript to address the reviewer comments. In particular, they have re-processed the cryo-EM data, performed additional experiments and revised the text. In my opinion, all major points have been adequately addressed and the manuscript can be published. I only have a few minor points left that I noticed:

We thank the Reviewer for the positive comments on our revised manuscript and appreciate the recommendation to publish our study.

1.) Line 160: I think Loop 1 needs to be introduced, i.e. stated that it corresponds to the loop described in the previous sentence.

In the revised manuscript, we have now defined loop 1 in the preceding sentence (now in line 169):

“In the free PYROXD1 crystal structure¹⁹, a highly conserved loop within the FAD binding domain (residues Pro45^{PYROXD1}-Arg76^{PYROXD1}, hereafter referred to as Loop 1), which contains an invariant sequence motif (Motif 1), is well structured and positioned away from the FAD co-factor (**Fig. 3B, Extended data Fig. 4A**).“

2.) Line 167-169: This sentence ends abruptly and should be revised.

We thank the Reviewer for pointing us to this sentence. We have revised the sentence, which now reads (Page 6, Line 178-184):

“A long loop (residues Ala197^{PYROXD1}-Glu251^{PYROXD1}, hereafter referred to as Loop 2), which connects the NAD(P)H binding and C-terminal domains, is structurally disordered in the NAD(P)H-free crystal structure of PYROXD1¹⁹. However, in the RTCB-PYROXD1 complex, segments of Loop 2 become partially ordered through interactions with the restructured NAD(P)H binding domain (**Fig. 3B, right, and Extended Data Fig. 4E-G**)¹⁹. This positions a conserved motif (Motif 2) within Loop 2 (**Extended data Fig. 4D**) such that the side chain of Trp239^{PYROXD1} contacts the flavin ring of FADH⁻ via side-on aromatic stacking (**Fig. 3B and Extended data Fig. 4F**).

3.) Extended Data Figure 2: Please check the particle numbers, they are identical at several steps.

We thank the Reviewer for pointing out this error. Extended Data Figure 2 has been revised to display the correct number of particles at each processing step.